# Steady-state forms of channel profiles shaped by debris-flow and fluvial processes

Luke A. McGuire[1], Scott W. McCoy[2], Odin Marc[3], William Struble[1], and Katherine R. Barnhart[4]

[1]The University of Arizona, Department of Geosciences, Gould-Simpson Building, 1040 East Fourth Street, Tucson, Arizona 85721, USA
[2]Department of Geological Sciences and Engineering, University of Nevada, Reno, Reno, NV, 89557, USA
[3]Géosciences Environnement Toulouse (GET), UMR 5563, CNRS/IRD/CNES/UPS, Observatoire Midi-Pyrénées, Toulouse, France
[4]U.S. Geological Survey, P.O. Box 25046, MS 966, Denver, Colorado 80225, USA

**Correspondence:** Luke McGuire (lmcguire@arizona.edu)

**Abstract.** Debris flows regularly traverse bedrock channels that dissect steep landscapes, but our understanding of bedrock erosion by debris flows and their impact on steepland morphology is still rudimentary. Quantitative models of steep bedrock channel networks are based on geomorphic transport laws designed to represent erosion by water-dominated flows. To quantify the impact of debris-flow erosion on steep channel network form, it is first necessary to develop methods to estimate spatial variations in bulk debris-flow properties (e.g., flow depth, velocity) throughout the channel network that can be integrated into landscape evolution models. Here, we propose and evaluate two methods to estimate spatial variations in bulk debris-flow properties along the length of a channel profile. We incorporate both methods into a model designed to simulate the evolution of longitudinal channel profiles that evolve in response to debris-flow and fluvial processes. To explore this model framework, we propose a general family of debris-flow erosion laws where erosion rate is a function of debris-flow depth and channel slope. Model results indicate that erosion by debris flows can explain the occurrence of a scaling break in the slope-area curve at low drainage areas and that upper-network channel morphology may be useful for inferring catchment-averaged erosion rates in quasi-steady landscapes. Validating specific forms of a debris-flow incision law, however, would require more detailed model-data comparisons in specific landscapes where input parameters and channel morphometry can be better constrained. Results improve our ability to interpret topographic signals within steep channel networks and identify observational targets critical for constraining a debris-flow incision law.

## 1 Introduction

Debris flows are effective at transporting coarse sediment (May, 2002; May and Gresswell, 2003) and eroding bedrock (McCoy et al., 2013; McCoy, 2015; Stock and Dietrich, 2006) in steep valleys and low-order channels where fluvial sediment transport may be inhibited by low runoff magnitudes and increases in thresholds for incipient sediment motion (e.g. Lamb et al., 2008;

Prancevic et al., 2014). The influence of debris-flow erosion on bedrock channel morphology, including longitudinal channel profiles (Montgomery and Foufoula-Georgiou, 1993), has been recognized for decades. Although debris flows traverse channel networks in many steep landscapes and are capable of eroding bedrock via abrasion and plucking (Hsu et al., 2008; Stock and Dietrich, 2006; McCoy et al., 2013), their relative importance over geologic time compared to other geomorphic processes and

the extent to which they affect landscape form at larger scales remains unclear. The spatial extent and magnitude of debris-flow erosion, for example, may be limited in terms of downstream extent due to a lack of mobility or erosive power on modest slopes. Additional work is needed to determine the controls on the magnitude of debris-flow erosion within different parts of the drainage network and the ensuing implications for landscape form (Whipple et al., 2013).

Topographic signatures of geomorphic processes, which we define as quantitative connections between processes and the

morphology of a landform, can be used to infer the presence and rates of geomorphic processes from topographic data (e.g. Tucker and Bras, 1998). Bedrock channels dominated by fluvial erosion develop longitudinal profiles described by a power law relationship between slope and drainage area (Flint, 1974; Morisawa, 1962; Hack, 1957). A deviation from this power-law scaling relationship at small drainage areas (Fig. 1) has been interpreted as a topographic signature of debris-flow incision (Montgomery and Foufoula-Georgiou, 1993; Seidl et al., 1992; Sklar and Dietrich, 1998; Stock and Dietrich, 2006, 2003).

Stock and Dietrich (2003) found that the shape of longitudinal channel profiles that experience both debris flow and fluvial erosion can be described by a family of curves with the general form

$$S = \frac{S_{df}}{1 + a_1 A^{a_2}} \tag{1}$$

where $A$ denotes upstream drainage area, and $S_{df}$, $a_1$, and $a_2$ are empirical coefficients (Fig. 1). Here, we use the term channel in a general sense to refer to an axis of concentrated erosion along valley bottoms, but which may or may not reside within

banks made of deposited sediment. The coefficient $S_{df}$ is related to the slope that the channel approaches at low drainage areas, $a_2$ controls the power-law relationship between slope and drainage area when $A$ is large, and $a_1$ (which has units of $1/(\text{length}^2)^{a_2}$) controls the sharpness of the transition from fluvial power-law scaling at large drainage areas to relatively constant slopes in debris-flow-dominated reaches at smaller drainage areas. The above expression for channel slope can be rewritten as

$$S = \frac{S_{df}}{1 + (A/A_{df})^{a_2}} \tag{2}$$

which is advantageous because $A_{df}$ has units of $\text{length}^2$ and can be interpreted as the drainage area at which the slope-area relationship transitions from a near-constant slope with decreasing drainage area to the standard power law relationship expected in the fluvial network. In this sense, $A_{df}$ provides one metric for identifying the transition between the debris-flow domain, with a characteristic gradient tending towards $S_{df}$, and the fluvial process domain.

Past work demonstrates that the length of the channel network upstream of the debris-flow to fluvial transition zone, which we roughly associate with $A_{df}$, increases with erosion rate in two landscapes where debris flows are known to regularly traverse steep channels, namely the San Gabriel Mountains (DiBiase et al., 2012) and the Oregon Coast Range (Penserini et al., 2017). These results from DiBiase et al. (2012) and Penserini et al. (2017) are consistent with the conceptual model proposed by

Stock and Dietrich (2003) where the transition from a nearly linear debris-flow dominated long-profile to a concave-up fluvial-dominated long-profile migrates out to larger drainage areas as the rock uplift rate increases. Therefore, there is support for the idea that steady state bedrock channels eroded by debris flows not only have a unique morphology (or topographic signature) that distinguishes them from purely fluvial channels, but they may also record tectonic information.

These findings underscore the need to develop a quantitative framework that can be used to explore topographic signatures generated by debris-flow erosion, assess the sensitivity of topographic signatures to climatic and tectonic forcing, and ultimately interpret these signatures to gain process-based insights about the evolution of steep landscapes. In particular, there is a need to understand the relative importance of fluvial and debris-flow processes in setting the location and form of the morphologic transition associated with $A_{df}$ (Fig. 1). For example, the location of this transition may change with rock uplift rate due to the dynamics of fluvial erosion alone because channel steepness can vary nonlinearly with rock uplift rate as a result of relationships between runoff variability and fluvial erosion thresholds (e.g. DiBiase and Whipple, 2011; Lague, 2014). Debris-flow processes, in contrast, may exert a strong control on the location of the transition by setting the near-uniform slope that the channel approaches at small drainage areas and the sharpness of the transition between near-linear and concave-up channels.Quantifying the roles of debris-flow and fluvial processes on controlling the location of this transition, as approximated by $A_{df}$, would aid in determining the benefits and limitations of using the morphology of debris-flow-dominated channels as a proxy for erosion rate in quasi-steady landscapes (Penserini et al., 2017).

In landscape evolution models, fluvial erosion is modeled based on empirical relationships between local terrain attributes such as slope and drainage area, readily computed from a digital elevation model (DEM) (Tucker and Bras, 1998). However, incorporation of a debris-flow incision law into this type of local framework is challenging owing to the nonlocal controls on debris-flow erosion including initiation conditions, non-steady flow velocity and the finite and variable runout distance of discrete flows. For example, the frequency at which discrete flows traverse different parts of the landscape has been shown to be a key factor in their ability to sculpt topography (Shelef and Hilley, 2016). We are aware of only one attempt to model the effects of debris-flow erosion over geologic timescales (Stock and Dietrich, 2006) aimed at reproducing channel profiles with the characteristic change in slope-area scaling observed in natural environments (Montgomery and Foufoula-Georgiou, 1993; Stock and Dietrich, 2003). Stock and Dietrich (2006) coupled empirical relationships for debris-flow properties with a debris-flow incision law based on inertial stress, but emphasized the need for improved methods to calculate spatially varying bulk debris-flow properties and additional studies to constrain a debris-flow incision law.

Several subsequent studies have improved our understanding of the grain-scale processes that control debris-flow incision rates and their relationship to bulk-flow properties that are more amenable to measurement and model simulation. Hsu et al. (2008, 2014) used physical experiments in a rotating drum to suggest that debris-flow erosion rates scale with bulk inertial stress, which can be cast as a function of flow shear rate, commonly assumed to be proportional to the ratio of depth-averaged flow velocity and flow depth. Similarly, McCoy et al. (2013) showed that while bed impact forces beneath erosive debris flows were broadly distributed and the result of discrete particle-bed impacts, these force distributions scaled with bulk flow properties such as flow depth. Following the erosion equation proposed by Sklar and Dietrich (2004) to account for bedrock erosion due to discrete particle impacts, McCoy (2012) used a series of discrete element simulations to show that erosion by

steady granular flows likely scales approximately linearly with flow depth and in a strongly nonlinear way with bed slope. The utility of these relationships in a landscape evolution model, however, requires tractable simulation of the spatial and temporal variability in the properties of individual debris flows (e.g. depth, velocity, shear rate) throughout the channel network and integration of the effects of numerous debris flow events on channel evolution over geologic time scales.

Here, we address this gap by developing a nonlocal modeling framework to predict the evolution of a 1d channel profile eroded by both fluvial and debris-flow processes. In this framework, the routing of debris flows down the channel profile as well as estimates of spatial and temporal variations in bulk debris-flow properties are represented either through a process-based model that relies on a set of partial differential equations (Iverson and Denlinger, 2001) or a reduced complexity approach motivated by Gorr et al. (2022) based on empirical relationships (Rickenmann, 1999). Calculations of spatial variations in debris-flow properties from these two methodologies provide a robust foundation for utilizing relationships between bulk debris-flow properties and particle impact forces (Hsu et al., 2008; McCoy et al., 2013) to estimate rates of bedrock incision by debris flows. We propose a general family of debris-flow erosion laws, with erosion rate being a function of debris-flow depth and channel slope, to illustrate how the proposed framework may be used to help constrain a debris-flow erosion law and to explore model sensitivity. We examine the extent to which different erosion laws are capable of reproducing the relationship between slope and drainage area, as captured by equation 2, that has been observed in steep, debris-flow prone landscapes and interpreted as a topographic signature of debris flows. The process-based and empirical routing approaches differ in their assumptions and complexity, and comparison yields insight into the relative importance of these differences as compared with the proposed form of a debris-flow erosion law. We finish with a sensitivity analysis to explore controls on $A_{df}$ and $S_{df}$, two metrics that capture basic aspects of debris-flow-dominated channel morphology.

## 2 Methods

### 2.1 Model framework

In the proposed 1d model framework, which is designed to simulate longitudinal channel profiles, the rate of change of elevation, $z$, with time, $t$, in a bedrock channel is driven by the rock uplift rate, $U$, fluvial erosion, $E_f$, and erosion by debris flows, $E_{df}$, according to:

$$\frac{\partial z}{\partial t} = U - E_f - E_{df}. \tag{3}$$

We solve equation 3 numerically on a one-dimensional grid with a uniform spacing of $\Delta x = 5$ m and use the standard explicit forward Euler method for time stepping. We chose a grid spacing of 5 m since it is small enough to resolve changes in the longitudinal channel profile and also large enough to keep model run times, which increase with decreasing grid spacing, manageable.

Fluvial erosion is computed using the threshold-stochastic stream power incision model presented by Lague (2014),

$$E_f = KA^{m_s}S^{n_s} \tag{4}$$

where $A$ denotes upstream drainage area as a function of distance, $x$, from the channel head, $S$ denotes slope, and $K$, and $m_s$ and $n_s$ are empirical parameters that depend on relationships between discharge and channel width, $w$, hydraulic geometry and discharge variability, and grain size. Here, we assume that drainage area varies with distance from the channel head according to $A = A_0 + 25x^{5/3}$, with $A_0 = 1000$ m$^2$ and the exponent of $5/3$ chosen to be consistent with a Hack's exponent of $3/5$. We assume channel width increases with drainage area as $w = k_w A^b$, where $k_w = 0.05$ and $b = 0.3$. Motivated by the geomorphic importance of debris flows in the San Gabriel Mountains (Lavé and Burbank, 2004), parameters related to channel geometry, including $k_w$ and $b$ that are related to width-area scaling (Fig. B1), and fluvial incision were selected based on DiBiase and Whipple (2011) and typical ranges reported by Lague (2014). These parameter choices result in $m_s = 1.4$ and $n_s = 2.33$. Complete details on parameter choices for the stream power model are given in Appendix A. Unless noted otherwise, parameters used for the fluvial erosion model are listed in Table G1.

We propose a general formulation that can be used to estimate the erosion rate attributable to debris flows, $E_{df}$, at a point on the landscape, given information about the bulk properties of the flow. In this work, we assume that bulk properties of a debris flow for a given landscape position, do not change. In other words, debris flow erosion is driven over time by repeatedly routing the same debris flow over the landscape. Motivated by observations that debris-flow erosion rates scale with bulk inertial stress (Hsu et al., 2008, 2014), a function of shear rate, and that grain-scale bed-impact force distributions scale with flow depth (McCoy et al., 2013), it is reasonable to postulate an erosion law that includes debris-flow depth and velocity. Since steady granular flows down inclined planes of increasing angles show a monotonically increasing relationship between slope angle and velocity (Silbert et al., 2001), slope may serve as a proxy for velocity. Here, we define $E_{df}$ as a function of channel slope and debris-flow depth, $h$. Debris-flow depth varies with position along the channel profile and with time throughout the course of a debris flow event. Letting $t_0$ denote the time a debris flow begins moving over a given location along the channel profile and $t_f$ the time when it has completely passed that location, then

$$E_{df} = k_{df} \int_{t_0}^{t_f} S^{\alpha} h^{\beta} \Phi dt \tag{5}$$

where the debris flow erodibility coefficient is defined as $k_{df} = \kappa_{df} F_{df}$, $\kappa_{df}$ is an empirical coefficient related to bedrock and flow properties (e.g. grain size), $F_{df}$ is a term quantifying the frequency of debris flow, $\Phi$ is a threshold factor that reflects a reduction in incision when the debris flow is close to rest, and $\alpha$ and $\beta$ are empirical exponents. We use this family of erosion laws to begin exploring the model framework we propose here. The model is designed in such a way that it would be straightforward to insert alternative erosion laws in the future. This formulation could also be extended to account for a distribution of representative debris flows with different properties given information about their relative recurrence. With the process-based debris-flow routing model (Section 2.2), we compute time-varying flow properties required to determine a debris-flow incision rate at each point along the channel profile using equation 5.

We also present a reduced-complexity routing algorithm, which closely follows the methodology presented by Gorr et al. (2022) to rapidly simulate debris-flow runout for hazard assessment purposes, to compute spatial variations in bulk debris-flow properties along a longitudinal channel profile (Section 2.3). In this approach, we use a set of empirical relationships that

relate debris-flow properties to topographic slope (Rickenmann, 1999) in order to estimate representative values for debris-flow depth, $h$, at each point along the debris-flow path as well as the time it takes for the debris flow to pass over that point, $t_f - t_0$, which we denote as $t_p$. In other words, $h$ varies along the channel profile but we neglect variations in $h$ that occur within individual debris-flow events (i.e. the rise and fall in flow depth as a debris flow passes over a point on the landscape). In this case, we employ a debris-flow erosion equation analogous to equation 5 that is simplified because $h$ is constant for a given channel location,

$$E_{df} = k_{df} t_p S^\alpha h^\beta \Theta \tag{6}$$

where $\Theta$ denotes a threshold factor that reflects a reduction in incision when the debris flow is close to rest. To assess the simplifying assumptions of this approach within the context of modeling the evolution of longitudinal channel profiles, we compare the morphology of modeled profiles using this reduced complexity algorithm with profiles generated using the process-based debris-flow routing model. This comparison is limited, as described in the following sections in more detail, to cases where downstream changes in debris-flow volume are assumed to be negligible. While we acknowledge this is not likely to be true in many natural settings (Santi et al., 2008; Santi and Morandi, 2013; Schürch et al., 2011), examining this end-member case allows for the most direct comparison between channel profiles produced by the model when using these two different debris-flow routing methods.

## 2.2 Estimating debris-flow incision with a process-based routing model

The initial step in computing the erosion rate attributable to debris flows is to determine the runout path of the debris flow as well as its bulk properties at different points along that path. The process-based debris-flow routing model is based on a set of conservation laws for mass and momentum within a depth-averaged framework. This particular model formulation was chosen because it provides sufficient complexity to enable exploration of the links between flow properties and the morphology of the resulting channel profile. The governing equations represent the flow of a two-component mixture, solids suspended in a Newtonian fluid (Iverson and Denlinger, 2001), in a rectangular channel with variable width (Vázquez-Cendón, 1999):

$$\frac{\partial h}{\partial t} + \frac{\partial (hv)}{\partial x} = -\frac{vh}{w}\frac{\partial w}{\partial x} \tag{7}$$

$$\frac{\partial (hv)}{\partial t} + \frac{\partial}{\partial x}\left(hv^2 + \frac{1}{2}g_z h^2\right) = g_x h - sgn(v)(1-\lambda)g_z h\phi(I) - \frac{2v\eta v_f}{\rho h} - \frac{v^2 h}{w}\frac{\partial w}{\partial x}. \tag{8}$$

Here, $w$ is the channel width, $h$ is flow depth, $v$ is velocity, $\phi(I)$ is the friction coefficient that depends on the inertial number (Jop et al., 2006), $I$, $g_x$ and $g_z$ denote components of gravity in the downslope and slope-normal directions, respectively, $\lambda = p_{bed}/\rho g_z h$ is the ratio of pore fluid pressure to total basal normal stress, $v_f = 0.5$ is the fluid volume fraction, and $\eta$ is the viscosity of the pore fluid. The first, second, and third source terms on the right hand side of the momentum conservation equation (equation 8) account for variations in bed topography, frictional resistance associated with the solid phase of the flow,

and viscous resistance associated with the fluid phase, respectively. The remaining source terms in the mass and momentum equations account for variations in channel width.

We assume that the ratio of pore fluid pressure to total basal normal stress decays with time since the debris flow entered the model domain, $t$, according to

$$\lambda = \lambda_0 \left[ 1 - \mathrm{erfc}\left( \frac{2h}{\sqrt{4Dt}} \right) \right], \tag{9}$$

where $\lambda_0 = 0.9$ and $D$ is the pore fluid pressure diffusivity. This approximation is consistent with an initially high pore fluid pressure shortly following initiation and subsequent linear diffusion of pore fluid pressure (Iverson and Denlinger, 2001) over time.

The friction coefficient is a function of the inertial number (Jop et al., 2006),

$$\phi(I) = \mu_s + (\mu_2 - \mu_s)/(I_0/I + 1) \tag{10}$$

where $I = \dot{\gamma} D_{eff}/(P/\rho_s)^{0.5}$, with $P$ denoting the basal normal stress, $\dot{\gamma} = 2v/h$ is the shear rate, $\rho_s = 2600$ kg m$^{-3}$ is the density of sediment, $I_0 = 0.279$ is a constant, $\mu_s = 0.382$, $\mu_2 = 0.644$, and $D_{eff}$ is a characteristic particle diameter. In this formulation, the friction coefficient increases with the inertial number and approaches $\mu_2$ when $I$ is large.

The governing equations are solved numerically on a grid with uniform spacing. We use a first-order, shock-capturing finite volume method with a Harten-Lax-van Leer-Contact (HLLC) approximate Riemann solver (Toro, 2009) to compute the fluxes across each grid cell boundary (McGuire et al., 2016, 2017). Source terms are treated separately with an explicit, first-order forward Euler method for time stepping.

Debris flows enter the domain through the upper boundary, which can be conceptualized as the channel head, and are routed down the channel profile. We define a series of 20 ghost cells above the uppermost grid cell that effectively extend the model domain for the purpose of initializing a debris flow. Elevations of each ghost cell are determined by assuming that the slopes of all ghost cells are equal to the slope at the uppermost grid cell. Debris flows are initiated from a static pile of debris defined on the ghost cells. This procedure provides some time for debris to begin to flow before it enters the model domain, similar to what might be expected for debris flows that initiate in a colluvial hollow or gully upstream from a channel head. In nature, we expect debris-flow volume to vary with drainage area as sediment is entrained and deposited along the runout path (Santi and Morandi, 2013; Schürch et al., 2011; Santi et al., 2008), but incorporating this effect into the source terms of the process-based routing model is beyond the scope of this study. When using the process-based debris-flow routing model, we assume that debris-flow volume is fixed and does not change along the flow path, although we do explore the effects of spatial variations in debris-flow volume with the empirical routing approach described later. In addition, we perform a set of numerical experiments with the process-based routing model where we scale debris-flow frequency with drainage area to account for an increase in the total volume of sediment transported by debris flows as drainage area increases. Regardless of which routing approach is used, however, we do not explicitly account for rock mass incorporated into the debris flow originating from bedrock incision. This sediment volume would be negligible compared to the total debris-flow volume.

At each grid cell in the model domain (i.e. excluding ghost cells), the debris-flow incision rate is computed using equation 5 based on the time-varying values of debris-flow depth. More specifically, for a debris flow simulated over $k$ timesteps,

$$E_{df} = k_{df} S^\alpha \sum_{k=1}^{k=n} h_k^\beta \Phi \Delta t. \tag{11}$$

where $\Delta t$ denotes the time step used when solving the flow equations and we define the threshold factor, $\Phi$, as $\Phi = 1$ when $uh > 0.01 \text{ m}^2 \text{ s}^{-1}$ and $\Phi = 0$ otherwise (Fig. E1). To reduce computation time, the term $\sum_{k=1}^{k=n} h_k^\beta \Phi$ is not updated with each

time step in the landscape evolution model. Small changes in topography, such as may occur during a single time step of the landscape evolution model, will not substantially affect flow mobility or spatial variations in flow depth along the runout path. Instead, we only route a debris flow down the channel profile to update $\sum_{k=1}^{k=n} h_k^\beta \Phi$ in the calculation of $E_{df}$ whenever the channel slope has changed by $0.05$ or more at any grid cell since the last time a debris flow was routed. We do, however, update $E_{df}$ with every time step of the landscape evolution model to reflect changes in slope, $S$, since this requires little computation

time compared with debris-flow routing. This is one benefit of using slope as a proxy for velocity in the debris-flow erosion law.

## 2.3  Estimating debris-flow incision with an empirical routing model

We use a series of empirical relationships defined by Rickenmann (1999) to estimate representative values for debris-flow depth, $h$, and passage time, $t_p$, at each point along the channel profile based on spatially variable estimates of debris-flow

volume, $M$, debris-flow velocity, $v$, channel width, $w$, and topographic slope, $S$. We assume that debris flows initiate at or above the uppermost grid cell within the computational domain (i.e. the channel head), athough their overall volume may change along the channel profile. We determine the downstream extent of debris-flow runout by treating the debris flow as an idealized fluid with a prescribed yield strength, $\tau_y$, and assuming that debris-flow motion stops when shear stress at the base of the flow, $\tau = \rho_b g R_h \sin\theta$, falls below $\tau_y$ (Whipple and Dunne, 1992; Gorr et al., 2022). Here, $R_h = wh/(w + 2h)$ denotes

the hydraulic radius of the rectangular channel, $\theta$ is the channel slope angle, $g = 9.81 \text{ m s}^{-2}$ denotes gravitational acceleration and $\rho_b = 1800 \text{ kg m}^{-2}$ is the bulk density of the debris flow. In practice, we determine $R_h$, $t_p$, and $\tau$ everywhere in the model domain, determine the downstream extent of debris-flow runout, and then apply equation 6 to compute a non-zero value for $E_{df}$ only along the debris-flow travel path.

To begin, we specify debris-flow volume passing through each grid cell as a function of upstream drainage area ($A$) according

to $M = M_0 (10^{-6} \cdot A)^\gamma$ (Santi and Morandi, 2013). This formulation assumes that debris-flow volume increases downstream, reflecting entrainment of bed material or lateral inflow, but these volume changes can be neglected by setting $\gamma = 0$. Debris-flow volume may not increase monotonically along the runout path (Schürch et al., 2011), but the formulation proposed by Santi and Morandi (2013) provides a useful starting point for a general parameterization of downstream variations in debris-flow volume, especially since there are data from a range of geographic regions to fit such a relationship. For example, Santi and

Morandi (2013) demonstrate that the empirical coefficient, $M_0$, and exponent, $\gamma$, may vary considerably among landscapes. Santi and Morandi (2013) estimated $M_0 = 3358$ and $\gamma = 0.73$ using data throughout the western and southwestern United States, $M_0 = 10470$ and $\gamma = 0.62$ based on data from the Italian Alps, and $M_0 = 18770$ and $\gamma = 0.28$ using data from the

northwestern United States and southwestern Canada. Peak debris flow discharge can then be computed according to

$$Q = c_1 M^{c_2}, \tag{12}$$

where $c_1 = 0.135$ and $c_2 = 0.78$ are empirical coefficients (Rickenmann, 1999). Noting that $Q = wvh$ and using the relationship (Rickenmann, 1999)

$$v = \frac{1}{3\mu} \rho_b g h^2 S, \tag{13}$$

where $\mu$ denotes the dynamic viscosity of the flow, it is possible to solve for flow depth,

$$h = \left( \frac{3\mu c_1 M^{c_2}}{\rho_b g S w} \right)^{1/3}. \tag{14}$$

Using the relationships between channel width, $w$, and area, $A$, and debris-flow volume, $M$, and area, $A$, a representative flow depth for a given location along the channel profile can be written in terms of area and slope,

$$h = \left( \frac{3\mu c_1 (M_0(10^{-6} \cdot A)^\gamma)^{c_2}}{\rho_b g S k_w A^b} \right)^{1/3}. \tag{15}$$

We treat this flow depth as a representative value for each point in the drainage network but acknowledge that it may overestimate flow depth because equation 12 is used to estimate peak debris flow discharge. We further define the passage time of the debris flow as

$$t_p = \frac{M}{Q} = \frac{M_0(10^{-6} \cdot A)^\gamma}{c_1(M_0(10^{-6} \cdot A)^\gamma)^{c_2}}. \tag{16}$$

Finally, we define the threshold factor, $\Theta$, such that the debris-flow incision rate decreases as the flow approaches the end of its travel path and the shear stress at the base of the flow approaches the yield strength. Specifically,

$$\Theta = 1 - \frac{\tau_y}{\tau} \tag{17}$$

The debris-flow erosion rate can then be determined according to equation 6.

In this study, we fix all model parameters within a given simulation. As such, the channel profiles that develop can be thought of as reflecting the morphology of a channel shaped by the repeated impacts of a characteristic debris flow. Future studies could explore the effects of debris flows characterized by a distribution of parameters to better reflect natural variations in flow properties.

## 2.4 Simplified analytical solution

When using empirical relationships to determine flow properties along the debris-flow runout path, we can derive an approximate analytical solution for the slope of the upper, debris-flow-dominated reach of the channel at steady state. We begin by considering debris flow erosion as quantified by equations 6, 15, and 16. To arrive at an analytical solution, we then make several simplifying assumptions. First, we assume that fluvial erosion is negligible. Second, we assume that the channel is

sufficiently steep so that shear stress at the base of the debris flow greatly exceeds the yield strength (i.e., $\tau >> \tau_y$). This implies that debris flows always traverse the entire channel reach that we are modeling and that it is reasonable to neglect the entertainment threshold (i.e., $\Theta = 1$). Enforcing the condition that the channel profile has reached a steady state yields,

$$\frac{\partial z}{\partial t} = 0 = U - k_{df} S^\alpha \left( \frac{M_0 (10^{-6} \cdot A)^\gamma}{c_1 (M_0 (10^{-6} \cdot A)^\gamma)^{c_2}} \right) \left( \frac{3\mu c_1 (M_0 (10^{-6} \cdot A)^\gamma)^{c_2}}{\rho g S k_w A^b} \right)^{\beta/3} \tag{18}$$

Solving for slope as a function of drainage area, we obtain

$$S = \left( \frac{U}{k_{df} \lambda_1 \lambda_2 \lambda_3 \lambda_4} \right)^{\frac{1}{\alpha - \beta/3}} A^N \tag{19}$$

where

$$N = \frac{\beta/3 (b - \gamma c_2) + \gamma(c_2 - 1)}{\alpha - \beta/3} \tag{20}$$

and $\lambda_1, \lambda_2, \lambda_3, \lambda_4$ are given by

$$\lambda_1 = M_0^{c_2(\beta/3-1)+1} \tag{21}$$

$$\lambda_2 = 10^{-6\gamma(c_2(\beta/3-1)+1)} \tag{22}$$

$$\lambda_3 = c_1^{\beta/3-1} \tag{23}$$

$$\lambda_4 = \left( \frac{3\mu}{\rho g k_w} \right)^{\beta/3}. \tag{24}$$

From this analytical solution, we can see that slope increases with rock uplift rate and decreases with $k_{df}$, which would increase with bedrock erodibility and/or debris-flow frequency. The sensitivity of slope to changes in $U$ and $k_{df}$ is strongest when $\alpha - \beta/3$ is small and gets weaker as $\alpha - \beta/3$ increases. In addition, the relation between $S$ and $A$ will depend on $b$, $\beta$ and $\gamma$. The sign of the exponent, $N$, controls whether slope decreases or increases with drainage area, $A$. Without downstream increases in debris-flow volume (i.e., $\gamma = 0$), channel widening leads to debris flow thinning and therefore to steepening of the steady state channel slope with increasing drainage area. Slope decreases with drainage area when $N < 0$, or equivalently when

$$\beta/3(b - \gamma c_2) + \gamma(c_2 - 1) < 0. \tag{25}$$

Assuming $c_2$, the exponent in the power law relating peak debris-flow discharge to debris-flow volume, is less than 1 (Ricken-mann, 1999), then $\gamma(c_2 - 1)$ will always be negative. Therefore, $N$ will also always be negative when $b - \gamma c_2 < 0$, or $\gamma > 0.38$ given values of $b$ and $c_2$ used here, though this is a more restrictive condition than is necessary to ensure $N < 0$. Plots of $N$ as a function of $\alpha$ and $\beta$ for different values of $\gamma$ demonstrate that $N$ is less than zero in cases where $\gamma = 0.25$ and $\beta < 2$ (Fig. C1). In any case, we see that high $\alpha$ and low $\beta$ values promote limited variations of $S$ with $A$ by reducing the magnitude of $N$. However, a caveat is that the exact dependency of $S$ on $A$ may also be highly influenced by the relationship between $A$ and $k_{df}$, which accounts for debris-flow frequency (Stock and Dietrich, 2006). These dependency will not be formally explored in this first study but remain to be explored through future field or modelling studies. Furthermore, the dependence of $N$ on $b$ motivates field observation of the debris-flow channel width as a function of drainage area. In the following numerical experiments we aim to confirm the validity of this analytical solution and its consistency to a more process-based model. We further aim to determine the range of $\alpha$ and $\beta$ values that produce channel profiles consistent with the observational constraint on the relationship between slope and drainage area as summarized by equation 2.

## 2.5 Numerical experiments

Our numerical experiments have two goals, which are treated in turn. First, we assessed which erosion laws, as defined by different values of $\alpha$ and $\beta$, can reproduce the first-order characteristics of observed channel longitudinal profiles as well as how this may be affected by the choice of debris-flow routing model (i.e. process-based or empirical). Second, we performed a series of simulations aimed at understanding the sensitivity of $A_{df}$ and $S_{df}$ to model parameters.

### 2.5.1 A family of debris-flow incision laws

We explored model behavior for different values of $\alpha$ and $\beta$ in the family of incision laws described by equations 5 and 6 by comparing modeled, steady-state longitudinal profiles with those typical of debris-flow-dominated terrain. We did not try to recreate the channel morphology observed within specific watersheds or geographic regions. Instead, we aimed to provide some constraints on $\alpha$ and $\beta$ by identifying ranges for these two exponents that resulted in longitudinal channel profiles that are consistent with observed changes in the relationship between slope and contributing area in natural channels traversed by debris flows (Stock and Dietrich, 2003).

A landscape evolution model designed to simulate the evolution of channel longitudinal profiles in response to both debris flow and fluvial erosion should produce steady-state channel profiles that are well described by equations 1 and 2. Equation 1, which was formulated by Stock and Dietrich (2003) as part of an analysis of channel morphology across a range of geographic areas, suggests that channel slope increases or remains approximately constant as drainage area decreases. We performed an analysis of 31 channel longitudinal profiles in the San Gabriel Mountains, USA, to determine the frequency with which channel slope decreased as drainage area decreased. The San Gabriel Mountains were chosen for this analysis because some of our model parameter choices are based on previous studies in this mountain range and topography is in an approximate steady state (DiBiase et al., 2012). We extracted channel profiles for a subset of catchments with [10]Be catchment-averaged erosion rates (DiBiase et al., 2010), where we eliminated catchments with signs of disequilibrium such as knickpoints. In 30 of the

31 catchments, in which erosion rate varied widely from $< 0.1 \text{ mm/yr}$ to more than $1 \text{ mm/yr}$, slope increased or remained approximately constant as drainage area decreased. In one catchment, there was a difference of $0.03$ between the maximum slope along the channel profile and the top of channel profile (Fig. B2).

We therefore assessed model performance for different $\alpha$ and $\beta$ in two ways. First, we computed the $R^2$ associated with the best fit to equation 2. We allowed $S_{df}$, $A_{df}$, and $a_2$ to vary freely when fitting to equation 2. Second, we examined the difference between the maximum slope along the channel profile, $S_{max}$, and the slope at the channel head, $S_{ch}$. The second criteria focused on checking a basic morphologic property observed in natural channels, namely that channel slope generally increases or remains constant as drainage area decreases in quasi-steady-state landscapes.

We assessed performance of the landscape evolution model with different values of $\alpha$ and $\beta$ when using the process-based routing model and when using the empirical routing model. Using the process-based model, we performed a numerical experiment where we varied $\alpha$, $\beta$, pore pressure diffusivity, $D$, viscosity of the pore fluid , $\eta$, the friction parameter, $\mu_2$, the debris flow erodibility coefficient, $k_{df}$, and instantaneous fluvial erodibility coefficient , $k_e$ (Appendix A), within the ranges specified in Table G2. We allowed some variation in model parameters other than $\alpha$ and $\beta$ to ensure trends between $\alpha$, $\beta$, and model per-

formance metrics were not specific to a particular subset of the parameter space. We selected 500 parameter sets using a Latin Hypercube sampling strategy. We performed an analogous numerical experiment using the empirical routing model where we sampled 4000 different parameter sets with varying values of $\alpha$, $\beta$, instantaneous fluvial erodibility ($k_e$), debris flow erodibility ($k_{df}$), viscosity ($\mu$), yield strength ($\tau_y$), and debris-flow volume parameters $M_0$ and $\gamma$ within prescribed ranges (Table G3). We were able to perform a greater number of simulations using the empirical model because it is less computationally demanding.

All simulations began with an initial condition determined by the analytical solution for a steady-state fluvial channel, specifically

$$S = (U/K)^{1/n_s} A^{-m_s/n_s}. \tag{26}$$

Simulations ended once an approximate steady state had been reached, which typically took $10^6 - 10^7$ years.

### 2.5.2   Sensitivity analysis

We performed sensitivity analyses using both the process-based and empirical routing models to explore how the topographic signature of debris-flow incision is likely to be expressed in different settings. Motivated by the results of our numerical experiments to constrain $\alpha$ and $\beta$ and by insights from the simplified analytical solution, we set $\alpha = 6$ and $\beta = 1$ for the sensitivity analysis. The analytical solution for slope suggests that the exponent, $N$, that controls whether slope increases or decreases with drainage area, will be negative for $\gamma > 0.25$ and relatively low in absolute value for $\gamma = 0$ when $\alpha = 6$ and $\beta = 1$.

Therefore, we focused on this combination of $\alpha$ and $\beta$ as it is likely to yield results that are consistent with observations as summarized by equation 2. We focused, in particular, on understanding relationships between model parameters and resulting longitudinal profile form as quantified by $A_{df}$ and $S_{df}$ in steady-state longitudinal profiles.

    To perform the sensitivity analysis with the process-based routing model, we used a Latin Hypercube sampling strategy to select 1500 sets of parameters where instantaneous fluvial erodibility, $k_e$, debris flow erodibility, $k_{df}$, viscosity of the pore

fluid, $\eta$, pore fluid diffusivity, $D$, the friction factor, $\mu_2$, and rock uplift rate, $U$ varied within the ranges defined in Table G4. The sensitivity analysis using the empirical routing approach was analogous, but we were able to perform a greater number of simulations. We used a Latin Hypercube sampling strategy to select $4000$ sets of parameters where instantaneous fluvial erodibility, $k_e$, debris flow erodibility, $k_{df}$, viscosity, $\mu$, yield strength, $\tau_y$, rock uplift rate, $U$, and debris-flow volume parameters $M_0$ and $\gamma$ varied within the ranges defined in Table G5.

We performed a qualitative sensitivity analysis by visually examining model output using colored scatter plots and also performed a quantitative global sensitivity analysis using the PAWN method (Pianosi and Wagener, 2015) as implemented with the SALib Python package (Herman and Usher, 2017). The PAWN method is a density-based global sensitivity analysis method. The output of a PAWN sensitivity analysis consists of a sensitivity index for each input variable that summarizes its relative contribution to uncertainty in the output. The sensitivity index varies from $0$ to $1$, with greater values indicating a greater

relative importance of the parameter. By comparing the magnitudes of the sensitivity indices for different input parameters, we were able to rank them in terms of relative importance. We separately assessed sensitivity to each of two model outputs, $A_{df}$ and $S_{df}$, since these two metrics summarize basic morphologic information about steady state channel profiles. We performed PAWN sensitivity analyses separately for models that employ the process based and empirical routing approaches. The PAWN sensitivity analysis allowed us to rank input variables in terms of their relative importance for determining $A_{df}$ and $S_{df}$.

## 3 Results

### 3.1 Constraints on a debris-flow incision law

#### 3.1.1 Process-based routing model

At large drainage areas, modeled profiles exhibit a power law scaling between slope and drainage area that is expected based on the fluvial incision law (Figs. 2, 3). The $R^2$ value associated with a fit to equation 2 was greater than $0.95$ for approximately

$93\%$ of the modeled profiles. At low drainage areas, however, all parameter combinations produced channel profiles where slope began to decrease as drainage area decreased. In other words, the difference between the maximum slope, $S_{max}$, along the channel profile and the slope at the channel head, $S_{ch}$, was positive and regularly exceeded $0.2$ in cases where $\beta > 2$ (Fig. 2). This decrease in slope at low drainage areas, particularly a decrease in magnitude of more than $0.05$, is inconsistent with equation 2 and observations (Fig. B2) that indicate slope continues to increase or remain approximately constant as drainage

area decreases. Differences between $S_{max}$ and $S_{ch}$ decreased rapidly as $\alpha/\beta$ increased (Fig. 2). An increase in slope with drainage area near the channel head, however, is not an inevitable consequence of using the process-based routing model (Appexndix D).

#### 3.1.2 Empirical routing model

Modeled profiles exhibited the expected power law scaling between slope and drainage area at large drainage areas where

fluvial incision dominated debris-flow incision. The coefficient of determination ($R^2$) value associated with a fit to equation 2

was greater than $0.95$ for all modeled profiles. As with results obtained using the process-based routing model, some parameter combinations produced channel profiles where the maximum channel slope was not observed at the channel head (Fig. 3). Differences between $S_{max}$ and $S_{ch}$ are minor when $\alpha = 6$ and $\beta = 1$, but become more substantial as $\alpha$ decreases and/or as $\beta$ increases (Fig. 3).

More generally, the extent to which modeled channel profiles exhibit a decrease in slope at small drainage areas depends on $\alpha$, $\beta$, and the exponent $\gamma$ that controls the relationship between debris-flow volume and drainage area (Fig. 4). In cases where $\gamma < 0.25$, numerous combinations of $\alpha$ and $\beta$ lead to decreases in slope at low drainage areas. For any choice of $2 \leq \alpha \leq 8$ and $\beta \approx 1.5$ or less, $S_{max} - S_{ch}$ was always less than $0.05$ (Fig. 4). For cases where $\beta > 2$, $\alpha$ needed to be approximately 5 or greater to maintain $S_{max} - S_{ch} < 0.05$ for all values of $\gamma$. Differences between $S_{max}$ and $S_{ch}$ increase as $\beta$ increases and/or as $\alpha$ decreases. We were unable to directly explore the effects of spatial variations in debris-flow volume using the process-based routing model, where we neglect changes in debris-flow volume along the flow path (i.e. $\gamma = 0$).

### 3.2 Steady-state forms of channel profiles

#### 3.2.1 Process-based routing model

Two defining characteristics of the simulated steady-state channel profiles, the near-constant slope that they approach near the channel head and the minimum drainage area at which there is a power law scaling between slope and drainage area, can be summarized by the two metrics: $S_{df}$ and $A_{df}$. Results of the sensitivity analysis demonstrate that neither $A_{df}$ nor $S_{df}$ are particularly sensitive to parameters that primarily affect flow mobility, including viscosity of the pore fluid ($\eta$), friction parameters ($\mu_2$), and pore fluid pressure diffusivity ($D$) (Figs. 5, 6, Table 2). Rather, $A_{df}$ is most sensitive to the instantaneous fluvial erodibility ($k_e$), debris-flow erodibility ($k_{df}$), and rock uplift rate ($U$) whereas $S_{df}$ is controlled predominantly by $k_{df}$ and $U$ (Table 2).

The sensitivity of steady-state long-channel profiles to changes in rock uplift rate leads to power law relationships between $A_{df}$ and $U$ and between $S_{df}$ and $U$ (Fig. 7). By randomly sampling model parameters, including rock uplift rate, within prescribed ranges, we assume that none are correlated with each other. However, this is unlikely in natural landscapes and correlations are expected. For example, we may expect that debris-flow frequency, $F_{df}$, increases with rock uplift rate. The consequences of such a correlation can be seen by examining the effects of $k_{df}$ on $A_{df}$ and $S_{df}$ for a given rock uplift rate (Fig. 7). Increases in $k_{df}$, for a given rock uplift rate, lead to increases in $A_{df}$ and decreases in $S_{df}$. A correlation between rock uplift rate and $k_{df}$ would therefore be likely to influence the fit between $S_{df}$ and $A_{df}$.

#### 3.2.2 Empirical routing model

Simulations indicate that $S_{df}$ is most sensitive to changes in $\gamma$, which controls the relationship between debris-flow volume and drainage area, and $k_{df}$, which is related to debris-flow frequency and bedrock erodibility followed by rock uplift rate (Figs. 8, 9, Table 3). Typical values of $S_{df}$ decrease with $k_{df}$ and increase with $\gamma$ and $U$ but are not strongly controlled by $k_e$, $\tau_y$, $\mu$, and $M_0$ (Table 3). The area at which there is a transition to fluvial power law scaling between slope and area, $A_{df}$, is most

sensitive to $\gamma$, $k_{df}$, $k_e$, $U$, and $k_{df}$ whereas it is relatively insensitive to $M_0$, $\mu$ and $\tau_y$ (Figs. 8, 9, Table 3). Mean values of $A_{df}$ tend to decrease strongly with $k_e$, increase with $k_{df}$, and decrease with $\gamma$. Parameters more directly related to the physical

properties of the debris flows, viscosity,$\mu$, and yield strength, $\tau_y$, had relatively minor control over $S_{df}$ and $A_{df}$ (Figs. 8, 9).

There is a power-law relationship between $A_{df}$ and rock uplift rate, $U$, although there is considerable scatter due to the wide range of parameter values included in the sensitivity analysis (Fig. 10). By randomly sampling the model parameters within prescribed ranges, we assume that none are correlated with each other. However, this is unlikely in natural landscapes and correlations are expected. For example we may expect that $F_{df}$ and/or $\gamma$ increase with rock uplift rate. Again, we explore the

consequence of such correlations by examining patterns in colored scatter plots that can help visualize the impact of $k_{df}$ on the relationship between $U$ and either $A_{df}$ or $S_{df}$ (Fig. 10). A considerable amount of scatter in the relationship between $U$ and $A_{df}$ appears to be attributable to variations in $\gamma$ and $k_{df}$, as expected from results of the PAWN sensitivity analysis (Table 3).

## 4 Discussion

### 4.1 Constraints on a geomorphic transport law for debris-flow incision

Results indicate that many members within the proposed family of debris-flow incision laws, as formulated by equations 5 and 6, produce channel profiles that are consistent with observations from natural landscapes (Figs. 2, 3). This is true within a wide range of the parameter space explored here, including for a range of $\gamma$ that covers the variability observed across several different geographic regions reported by Santi and Morandi (2013). Data and numerical experiments presented here are not capable of differentiating among these potential debris-flow incision laws, although cases where $\alpha < 3$ and/or $\beta > 2$ generally

performed poorly (Figs. 2, 3). Additional work is needed to formulate and test a debris-flow incision law, including incision laws not restricted to the form of equation 5 (e.g. Stock and Dietrich, 2006). Stock and Dietrich (2006), for example, present a debris-flow incision law based on inertial stress. Analogously, there are a range of exponents used in the generalized stream power incision law for fluvial erosion and work continues in an effort to constrain those exponents (e.g. Clubb et al., 2016; Turowski, 2018, 2021). Based on the extent to which key characteristics of long-channel profile morphology is affected by

$k_{df}$, $k_e$, $\gamma$, and $U$, ideal landscapes for testing a debris-flow incision law would be ones where there are constraints on these parameters (Figs. 5, 7, 8, 10).

Here, we assess different debris-flow incision laws based on their ability to reproduce a general pattern in slope-area data (i.e. equation 2) observed in debris-flow-dominated landscapes (e.g. (Fig. B2). To produce the observed steady-state morphology of debris-flow dominated long-profiles with a slope that is approximately constant or slowly decreasing with $A$, examination of

the analytical solution for slope in the upper channel network given by equation 19 demonstrates that $\gamma > b/c_2$ is a sufficient, though more restrictive than necessary, condition to ensure that steady state slopes decrease as drainage area increases. For simulations presented here, $c_2 = 0.78$ and $b = 0.3$, which implies $\gamma > 0.38$. Examination of equation 20 shows that when $\gamma = 0.25$, $N$ is generally small in magnitude, particularly for greater values of $\alpha$ (Fig. C1). Results of numerical simulations are consistent with the analytical solution, with $\gamma > 0.25$ being sufficient to ensure that slope is approximately constant or

decreases as drainage area increases (Fig. 4). Equation 19 also demonstrates how the magnitude of the exponent, $N$, is

modulated by $\alpha$ and $\beta$ (Fig. C1). The analytical solution is consistent with numerical simulations using both the empirical and process-based routing models that show a general trend toward greater differences between maximum channel slope and slope at the channel head as $\beta$ increases and as $\alpha$ decreases (Figs. 2, 3). Since both numerical and analytical solutions to the model equations demonstrate how $b$ and $\gamma$ exert a strong control on determining the basic morphology of channel profiles in debris-flow dominated reaches near the channel head (i.e. the sign of $N$ in equation 19), additional model evaluation criteria beyond those proposed here would be needed to test a debris-flow incision law. One possibility would be to assess the ability of a debris-flow incision law to reproduce observed trends between erosion rate and $A_{df}$ or $a_1$, such as that observed by Penserini et al. (2017) in the Oregon Coast Range.

In general, the proposed empirical and process-based approaches for estimating bulk debris-flow properties along the channel profile do not appear to result in different model behavior (Figs. 2, 3, 6, 9). It is not possible to directly compare the longitudinal profiles produced by the two different routing models with the same values of $\alpha$ and $\beta$ because the parameters that determine debris-flow mobility are different among the two models. For example, the process-based model has no yield strength parameter, yet this parameter plays a key role in determining debris-flow runout in the empirical model. Although the flow depth, velocity, and passage time predicted by the two routing models will undoubtedly vary, these variations are not sufficient to alter the extent to which different values of $\alpha$ and $\beta$ produce modeled profiles that are consistent or inconsistent with observations of channel morphology in debris-flow-dominated terrain (Figs. 2, 3). Furthermore, results using the process-based and empirical routing approaches both highlight the sensitivity of $A_{df}$ and $S_{df}$ to rock uplift rate, $k_e$, $k_{df}$, and $\gamma$ relative to model parameters related to flow mobility (Figs. 6, 9). These similarities in model behavior are encouraging because the empirical routing approach provides a framework to estimate bulk debris-flow properties using quantities that can be computed from a digital elevation model, specifically upstream contributing area and slope, that could be used in future efforts to more efficiently explore alternative debris-flow incision laws.

## 4.2 Steady-state forms of longitudinal channel profiles

Model results help clarify the roles played by debris-flow and fluvial erosion processes in setting longitudinal profile form in the upper channel network. Changes in parameters related solely to fluvial erosion do not have a strong influence on $S_{df}$, which simulations demonstrate is primarily controlled by changes in debris-flow processes (Figs. 5, 8, 9). Specifically, increases in rock uplift rate, $U$, and $\gamma$, or decreases in $k_{df}$ promote increases in $S_{df}$, assuming all else is fixed. In contrast, $A_{df}$ is controlled by a combination of fluvial and debris-flow processes (Figs. 5, 6, 8, 9). On average, increases in the instantaneous fluvial erodibility lead to decreases in $A_{df}$ whereas increases in the debris-flow erodibility coefficient promote increases in $A_{df}$. If $S_{df}$ was a constant set, for example, by soil geotechnical properties related to slope stability, then $A_{df}$ could be estimated by the area at which the steady-state fluvial channel gradient reaches $S_{df}$. However, this type of threshold behavior of $S_{df}$ is not what we observe. Rather, simulations demonstrate that $S_{df}$ varies with $U$, $\gamma$ and $k_{df}$ and independent changes to either debris-flow incision processes or fluvial processes are sufficient to influence $A_{df}$. Thus, accounting for both debris-flow and fluvial processes is important to understand how the steep channel network will respond to changes in tectonic or climatic forcing.

Simulations, assuming $\alpha = 6$ and $\beta = 1$, indicate that debris flows may frequently traverse channel reaches at larger drainage areas without influencing longitudinal channel profile form in a substantial way. Debris flows routed with the empirical model traversed the entire model domain ($8\,\mathrm{km}^2$) in greater than $99\%$ of simulations, but the median $A_{df}$ was approximately $0.4\,\mathrm{km}^2$. A primary reason for this is the sensitivity of the debris-flow incision law to slope when $\alpha = 6$, which means that portions of the channel profile could regularly be traversed by debris flows but they would do little work to erode bedrock when the channel slope is modest. This result indicates that the presence of debris-flow activity in natural channels, as indicated by debris-flow deposits, may not be a reliable indicator of the importance of debris-flow incision. However, additional work is needed to constrain the relationship between slope and debris-flow incision rates as well as to explore the influence of debris flows mobilizing large-caliber sediment below $A_{df}$ that would otherwise shield the bed from subsequent fluvial erosion. An additional consequence of the nonlinear ($\alpha >> 1$) relationship between slope and debris-flow incision and the observation that the majority of debris flows remain mobile throughout the entire model domain, is that $A_{df}$ and $S_{df}$ are less sensitive to parameters related to debris-flow properties, namely viscosity and yield strength in the case of the empirical routing model (Figs. 9, 10) and viscosity of the pore fluid, friction parameters, and pore pressure diffusivity in the case of the process-based routing model (Figs. 5, 6). As long as debris flows are sufficiently mobile to traverse moderate slopes, these parameters primarily affect the debris-flow incision rate by changing debris-flow depth and/or passage time. Because both debris-flow depth and passage time are linearly related to the debris-flow incision rate when $\beta = 1$, a factor-of-two increase or decrease in flow depth or passage time would only require a relatively small adjustment in slope to compensate for the ability of debris flows to balance the imposed rock uplift rate at steady state. PAWN sensitivity analyses conducted using results of $4000$ simulations for cases where $\alpha = 6$ and $\beta = 2$ (Table F1) and $4000$ simulations where $\alpha = 4$ and $\beta = 1$ (Table F2) demonstrate this conclusion, namely $A_{df}$ and $S_{df}$ being less sensitive to parameters related to debris-flow properties, also holds for other combinations of $\alpha$ and $\beta$.

### 4.3 Tectonics from debris-flow processes and topography

Model results support previous observations indicating that the morphology of channel profiles in debris-flow-dominated landscapes may provide constraints on erosion rates in steady-state landscapes (Figs. 7, 10). Penserini et al. (2017) document an inverse relationship between $a_1$ and $E$ in the central Oregon Coast Range based on analyses of channel profiles in six watersheds with catchment-averaged erosion rates determined from cosmogenic radionuclide analysis. Cast in terms of the morphologic variables used here to describe channel profiles, specifically using $A_{df}$ in place of $a_1$, the results from Penserini et al. (2017) indicate that $A_{df}$ increases with $E$. Simulations confirm this pattern of increasing $A_{df}$ with $E$, but also highlight the importance of constraining relationships between $E$ and $k_{df}$ as well as $E$ and $\gamma$ in order for $A_{df}$ to serve as a proxy for erosion rates in an absolute sense (Figs. 7, 10). This indicates a need to prioritize constraining relationships between $E$ and debris-flow frequency and bed erodibility (which together control $k_{df}$) as well as $E$ and the rate at which debris-flow volume increases with drainage area (e.g. $\gamma$) rather than potential relationships between $E$ and debris-flow mobility parameters, such as viscosity and yield strength. Although not explored here, differences in $m_s$ and $n_s$, which vary among landscapes, are also likely to influence relationships between $E$ and $A_{df}$. Similarly, several landscape parameters assumed fixed in this study, such

as the area at the channel head, $A_0$, or the channel width scaling, $b$, may be influenced by debris-flow erosion and exert control over the long-term channel profile morphology. Analytical (i.e. equation 25) and numerical (Figs. 5, 8) model results indicate

that relationships between drainage area and debris-flow volume as well as drainage area and channel width, in particular, play key roles in determining the morphology of the debris-flow-dominated channels.

Simulations indicate increases in erosion rate, or equivalently rock uplift rate in a steady-state landscape, lead to an increase in $S_{df}$ (Figs. 7, 10). Interestingly, Penserini et al. (2017) found no systematic variation in $S_{df}$ with erosion rate in the Oregon Coast Range. However, the lack of any relationship between $S_{df}$ and erosion rate ($E$) in the Oregon Coast Range may result

from a correlation between $E$ and debris-flow frequency and/or $E$ and debris-flow volume. Simulations of steady-state channel profiles indicate that $S_{df}$ increases with rock uplift rate in cases where there is no relationship between rock uplift rate ($U$) and $k_{df}$ but imposing an increase in $k_{df}$ with $U$ would alter that relationship (Figs. 7, 10). Thus, two avenues are essential for improving our ability to use the morphology of debris-flow-dominated channels as a proxy for erosion rate, namely (1) extracting upland channel morphology in a larger number of catchments with constrained erosion rates, and (2) gathering

evidence on the interconnections of key parameters (Stock and Dietrich, 2006).

## 4.4 Model applications and limitations

We describe a model that provides a framework for exploring the effect of episodic debris flows on channel longitudinal profiles. The model also comes with several limitations. We assume that all debris flows initiate at the channel head, whereas debris-flow initiation locations in natural landscapes will be more varied. Past work highlights the role that network structure, specifically

the number of debris-flow initiation locations upstream from a given channel reach, may play in controlling channel form (Stock and Dietrich, 2006). Variations in the spatial distribution of debris-flow initiation locations within a watershed could be explored within the model presented here by prescribing debris-flow frequency as a function of drainage area (e.g. Fig. D1) or explicitly modeling multiple debris flows from different initiation locations. Additionally, the scaling between channel width and drainage area may differ in the upper portion of the channel network from previously reported relationships that are derived

from data at larger drainage areas where fluvial processes are dominant (DiBiase and Whipple, 2011). However, as additional data become available to better parameterize channel morphometry in steep landscapes (Neely and DiBiase, 2023), particularly at small drainage areas, new parameterizations can be exchanged with those presented here. Similarly, when using the empirical debris-flow routing model, we rely on relationships between debris-flow volume and drainage area that were derived using data collected primarily at drainage areas greater than 0.1 km$^2$ (Santi and Morandi, 2013). Lastly, the process-based debris-flow

routing model presented here assumes that debris-flow volume is constant and does not change along the travel path, although quantifying controls on sediment entrainment by debris flows and incorporating entertainment into process-based debris-flow routing models are areas of active research (Iverson, 2012; Iverson and Ouyang, 2015; McCoy et al., 2012; Haas and Woerkom, 2016). Advances in our understanding of how debris flows entrain sediment could allow for more detailed comparisons between empirical and process-based approaches to sediment bulking in the proposed landform evolution model.

The landform evolution model presented here may serve as a basis for future studies that aim to test or validate potential debris-flow incision laws, incorporate debris-flow incision into 2d landscape evolution models, or explore how the upper chan-

nel network responds to tectonic or climatic perturbations. The process-based routing model may be best suited for modeling 1d channel profiles where changes in flow volume can be neglected and debris-flow constituents are sufficiently well known to allow for estimates of the model parameters, thereby minimizing the number of numerical experiments needed to characterize model behavior. The empirical debris-flow routing algorithm provides an efficient framework for investigating the effects of different debris-flow bulking relationships and exploring large parameter spaces. It is also particularly promising for application in 2d landscape evolution models given its simplicity relative to process-based debris-flow routing models and its ability to connect slope and drainage area, which are readily available in nearly all landscape evolution models, with bulk debris-flow properties relevant to debris-flow incision.

## 5 Conclusions

We present a novel framework for incorporating erosion by debris flows into a model for channel profile evolution. We propose two methods to estimate debris-flow runout and bulk debris-flow properties (e.g. depth, velocity) throughout the channel network, one based on a process-based debris-flow routing model and the other based on an empirical routing approach. Combined with a geomorphic transport law describing the relationship between debris-flow depth, channel slope, and a debris-flow incision rate, we are able to quantify spatial variations in the debris-flow incision rate throughout the channel network. We explore the performance of a family of potential debris-flow incision laws by comparing the form of modeled longitudinal channel profiles with those typically observed in debris-flow-dominated landscapes. Results demonstrate that a debris-flow incision law based on flow depth, slope, and debris-flow passage time can reproduce the relationship between slope and drainage area that has been interpreted as a topographic signature of debris flows, given general constraints on empirical exponents that relate flow depth and local channel slope to the incision rate. Since a large subset of the proposed family of erosion laws is capable of reproducing this topographic signature of debris flows, additional criteria and more precise bounds on poorly constrained model parameters are needed to test and validate debris flow erosion laws. Simulations indicate that both $A_{df}$ and $S_{df}$ have potential to serve as a morphologic proxy for the catchment-averaged erosion rate. However, both $A_{df}$ and $S_{df}$ are sensitive to debris-flow frequency and debris-flow erodibility ($k_{df}$), and the empirical exponent characterizing how debris-flow volume increases with drainage area ($\gamma$), indicating that the utility of such a proxy would depend on the extent to which relationships between erosion rate, $k_{df}$, and $\gamma$ could be constrained. Results provide a general framework that can be used to test debris-flow incision laws and explore the relative importance of debris-flow versus fluvial processes in shaping channel profiles in steep landscapes. Results take initial steps toward the broader inclusion of bedrock erosion by debris flows into landscape evolution models and provide insights into the relationship between debris-flow processes and channel profile morphology in steep landscapes.

*Code and data availability.* Model code is available on HydroShare at http://www.hydroshare.org/resource/53dc6bada7d441179fae07df079fcd75 (McGuire et al., 2023).

## Appendix A:  Stochastic stream power incision model

The parameterization for the stochastic stream power model is not tuned to any particular landscape or geographic region, but relies on values and relationships that are based on typical values reported by Lague (2014) for high discharge settings and by DiBiase and Whipple (2011) for the San Gabriel Mountains. Given the parameters listed in Table G1, we follow Lague (2014) and compute the critical shear stress for bed load transport, $\tau_c$ according to $\tau_c = 0.045g(\rho_s - \rho_w)D_{eff}$, with $D_{eff} = 0.09$ m the effective grain size. The stochastic-threshold prediction for the slope exponent in the fluvial incision law, $n_s$, is given by $n_s = \beta_f/\alpha_f(k+1)/(1-\omega_s)$, where $\beta_f = 0.7$ is the slope exponent in the hydraulic friction law, $\alpha_f = 0.6$ is the discharge exponent in the hydraulic friction law, $\omega_s$ is the at-a-station width scaling exponent, and $k = 0.5$ is the discharge variability coefficient. The prediction for the area exponent is given by $m_s = (c-b)(k+1)/(1-\omega_s)$, where $c = 1$ is the mean discharge-area scaling exponent and $b = 0.3$ is the width-area scaling coefficient (Lague, 2014). Lastly, the fluvial erodibility coefficient is calculated as (Lague, 2014)

$$K = k_s k_{wq}^{-\alpha n_s/\beta} R_c^{m_s/c} \tag{A1}$$

where $R_c = 0.28$ m denotes the mean annual runoff, $k_{wq} = R_c^{b/c}/k_w$ denotes a width factor, and

$$k_s = \left( \frac{a\alpha_f(1-\omega_s)k_t^{n_s/\beta_f}k^{k+1}\Gamma(k+1)^{-1}}{(k+1)(k+1-a\alpha_f(1-\omega_s))} \right) \tau_c^{a-n_s/\beta_f} k_e. \tag{A2}$$

Here, $k_t = g\rho_w N^{3/5}$, $N = 0.05$ denotes the Manning friction coefficient, and $a = 1.5$ is a shear stress exponent. The rate of fluvial erosion is then computed according to $E_f = KA^{m_s}S^{n_s}$.

## Appendix B:  San Gabriel Mountains channel morphology

Scaling relationships that relate drainage area and channel width are often derived from data that includes drainage areas greater than those modeled in this study. Recall that in this study we use the term channel to broadly refer to a concentrated axis of erosion along valley bottoms. Since debris flows initiate and traverse channels at low drainage areas, we quantified channel width at drainage areas less than 3 km$^2$ in the San Gabriel Mountains. More specifically, we focused on a region in the San Gabriel Mountains burned by the 2016 Fish Fire. A series of rainstorms in the first year following the fire led to runoff-generated debris flows that scoured many low-order channels to bedrock (Rengers et al., 2021; Tang et al., 2019). A post-event DEM derived from airborne lidar provided an opportunity to quantify channel width in this location. We estimated channel width by visually examining cross-channel profiles and identifying channel banks or distinct breaks in slope that indicated a shift from the hillslope to channel. Data indicate that channel width increases as a power law function of drainage area with an exponent of approximately 0.28 (Fig  B1). These data provide support for the width-area scaling used here, but we acknowledge that better characterization of the morphometry of debris-flow-dominated channels would improve the utility of landform evolution models that represent steep, low-order channels.

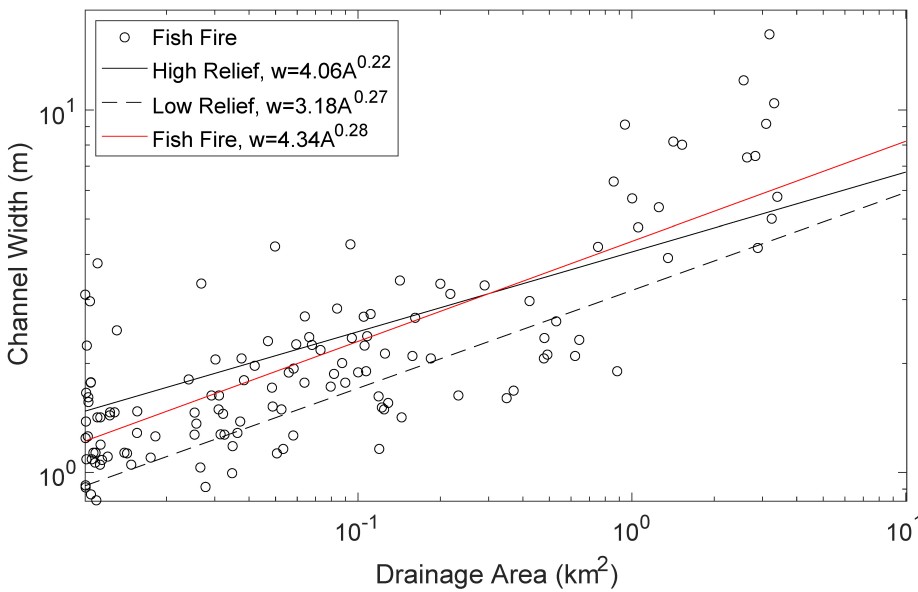

**Figure B1.** Estimates of channel width, estimated from a high resolution lidar-derived digital elevation model, as a function of drainage area for a portion of the San Gabriel Mountains, USA. The area burned in the 2016 Fish Fire and experienced a series of debris flows during the first rainy season following the fire that scoured valleys and channels to bedrock in many places. For comparison, relationships between channel width and drainage area as determined by DiBiase et al. (2012) are also shown.

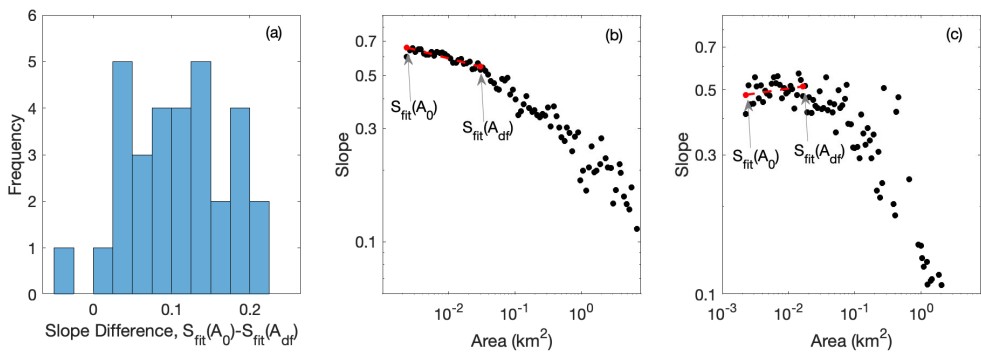

**Figure B2.** We fit a line, $S_{fit}$, to binned slope-area data for areas between $A_0$ and $A_{df}$. (a) Based on analysis of 31 channel profiles in the San Gabriel Mountains, differences between $S_{fit}(A_{df})$ and $S_{fit}(A_0)$ range from approximately 0.05 to 0.2. (b) Example of binned slope-area data for a channel profile where slope increases with decreasing drainage area from $A_{df}$ to $A_0$. (c) Example of binned slope area data where slope decreases slightly as drainage area decreases from $A_{df}$ to $A_0$.

## Appendix C:  Analytical solution

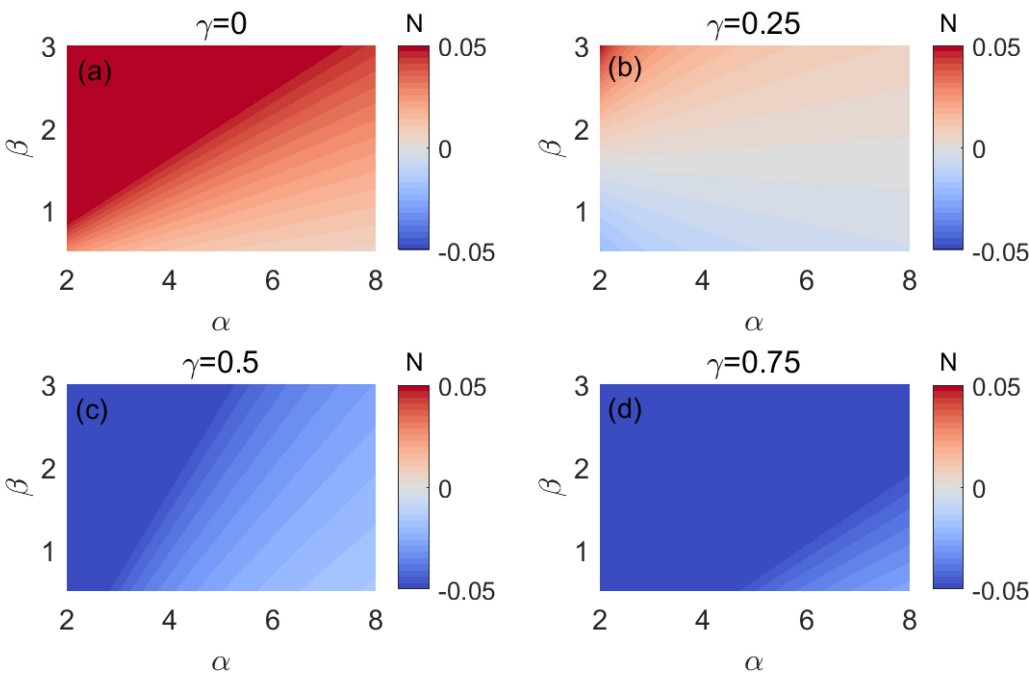

**Figure C1.** The analytical solution for steady state channel slope indicates that slope is a power law function of drainage area with an exponent, $N$, given by equation 20. Slope increases with drainage area when $N > 0$ and decreases with drainage area when $N < 0$. The above plots show how the magnitude and sign of $N$ vary with the two exponents in the debris-flow incision law, $\alpha$ and $\beta$, for increasing values of $\gamma$, the debris-flow volume-area scaling exponent, from (a) $\gamma = 0$, (b) $\gamma = 0.25$, $\gamma = 0.5$, and $\gamma = 0.75$.

## Appendix D:  Spatially variable debris-flow frequency

The process-based routing model does not directly account for downstream changes in debris-flow volume. When using the empirical routing model, for example, we prescribe debris-flow volume as a function of drainage area according to $M = M_0(10^{-6} \cdot A)^\gamma$. However, we can scale debris-flow frequency with drainage area in a way parameterizes an overall increase in the debris-flow volume as drainage area increases. For a basic illustration of this parameterization and its effects on the model solution, we performed a series of simulations where we scale $k_{df}$ by a factor of $1000(10^{-6} \cdot A)^\gamma / M_0$. This parameterization leads to an increase in the total sediment transported by debris flows as a function of drainage area that is consistent with the way in which debris flow volume increases downstream when using the empirical routing model. Results are summarized in Figure  D1 and lead to patterns that are qualitatively consistent with those obtained when parameterizing downstream increases in debris-flow volume with the empirical routing model (Fig.  9d).

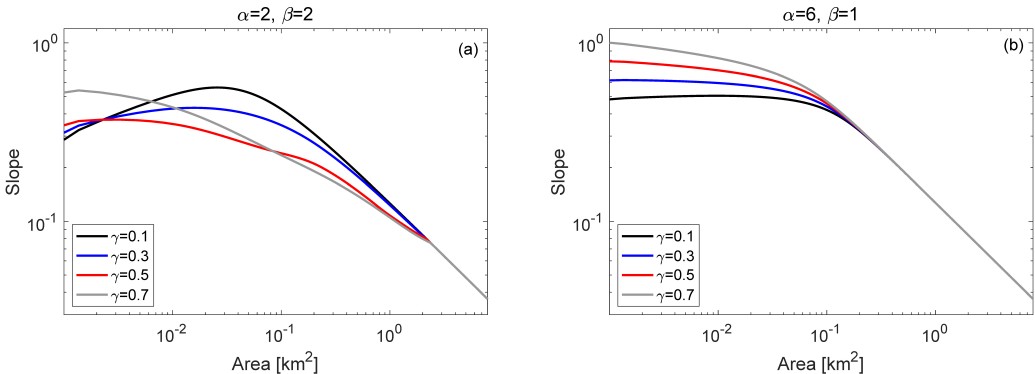

**Figure D1.** Steady state channel profiles using the process-based routing model where we parameterize an increase in debris-flow frequency with drainage area, $A$, by scaling $k_{df}$ by $1000 * (10^{-6} \cdot * A)^{\gamma} / M_0$. This parameterization leads to an increase in the total sediment transported by debris flows as a function of drainage area that is consistent with the way in which debris flow volume increases downstream when using the empirical routing model. All other parameters are fixed: $k_{df} = 5 \cdot 10^{-4}$; $k_e = 5 \cdot 10^{-14}$; $\eta = 60$; $\mu_2 = 0.67$; $\mu = 5 \cdot 10^{-6}$.

## Appendix E: Process-based debris-flow routing model

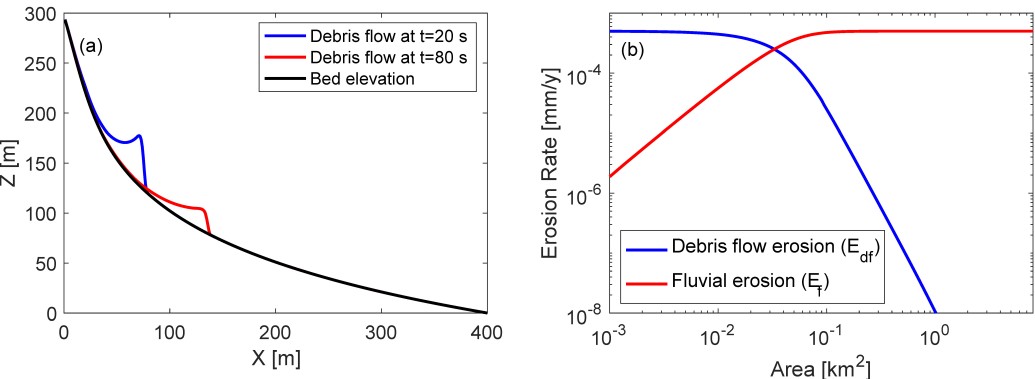

**Figure E1.** (a) A debris flow being routed down a channel profile using the process-based flow routing model. Flow depth is multiplied by a factor of 50 for display purposes. (b) The debris flow erosion rate varies spatially due to differences in slope and debris flow depth, including differences in temporal changes in flow depth as it passes over each point. The fluvial erosion rate is computed based on slope and drainage area at each point.

## Appendix F: PAWN Sensitivity Analysis

Here, we report results for the PAWN sensitivity analysis when using erosion laws with $\alpha = 6$ and $\beta = 2$ (Table F1) as well as $\alpha = 4$ and $\beta = 1$ (Table F2).

**Table F1.** PAWN Sensitivity Indices: Empirical Model, $\alpha = 6$, $\beta = 2$

| Parameter | Definition | Range | Sensitivity Index ($A_{df}$) | Sensitivity Index ($S_{df}$) |
|---|---|---|---|---|
| $M_0$ | Volume-area scaling coefficient | $1000 - 5000$ | 0.06 | 0.07 |
| $\gamma$ | Volume-area scaling exponent | $0 - 1$ | 0.35 | 0.36 |
| $\mu$ | Dynamic viscosity | $100 - 1000$ | 0.07 | 0.09 |
| $\tau_y$ | Yield strength | $100 - 600$ | 0.05 | 0.04 |
| $k_{df}$ | Debris flow erodibility coefficient | $5 \cdot 10^{-5} - 10^{-3}$ | 0.11 | 0.13 |
| $k_e$ | Instantaneous fluvial erodibility | $2 \cdot 10^{-14} - 8 \cdot 10^{-14}$ | 0.16 | 0.04 |
| $U$ | Rock uplift rate | $0.1 - 1$ | 0.13 | 0.1 |

**Table F2.** PAWN Sensitivity Indices: Empirical Model, $\alpha = 4$, $\beta = 1$

| Parameter | Definition | Range | Sensitivity Index ($A_{df}$) | Sensitivity Index ($S_{df}$) |
|---|---|---|---|---|
| $M_0$ | Volume-area scaling coefficient | $1000 - 5000$ | 0.07 | 0.08 |
| $\gamma$ | Volume-area scaling exponent | $0 - 1$ | 0.33 | 0.26 |
| $\mu$ | Dynamic viscosity | $100 - 1000$ | 0.05 | 0.06 |
| $\tau_y$ | Yield strength | $100 - 600$ | 0.05 | 0.06 |
| $k_{df}$ | Debris flow erodibility coefficient | $5 \cdot 10^{-5} - 10^{-3}$ | 0.14 | 0.20 |
| $k_e$ | Instantaneous fluvial erodibility | $2 \cdot 10^{-14} - 8 \cdot 10^{-14}$ | 0.16 | 0.05 |
| $U$ | Rock uplift rate | $0.1 - 1$ | 0.09 | 0.14 |

## Appendix G:  Model Parameters

Tables below provide details on the value or range of model parameters used in different numerical experiments.

**Table G1.** Stochastic Stream Power Model Parameters

| Symbol | Definition | Value | Unit | Basis for Value |
|--------|-----------|-------|------|-----------------|
| $\rho_w$ | Density of water | 1000 | $\text{kg m}^{-3}$ | |
| $\rho_s$ | Density of sediment | 2600 | $\text{kg m}^{-3}$ | |
| $D_{eff}$ | Effective grain size | 0.09 | m | DiBiase et al. (2011) |
| $b$ | Width-area scaling exponent | 0.3 | | Lague (2014) |
| $k_w$ | Width-area scaling coefficient | 0.05 | $\text{m}^{1/2b}$ | |
| $a$ | Shear stress exponent | 1.5 | | Lague (2014) |
| $c$ | Mean discharge-area scaling exponent | 1 | | Lague (2014) |
| $k$ | Discharge variability coefficient | 0.5 | | DiBiase et al. (2011) |
| $\omega_s$ | At a station width scaling exponent | 0.25 | | DiBiase et al. (2011) |
| $\alpha_f$ | Discharge exponent in hydraulic friction law | 0.6 | | Lague (2014) |
| $\beta_f$ | Slope exponent in hydraulic friction law | 0.7 | | Lague (2014) |
| $R_c$ | Mean annual runoff | 0.28 | m | DiBiase et al. (2011) |
| $N$ | Manning friction coefficient | 0.05 | $\text{s m}^{-1/3}$ | |

**Table G2.** Parameters used when running numerical experiments with the process-based routing model to constrain $\alpha$ and $\beta$

| Symbol | Definition | Value | Unit |
|--------|-----------|-------|------|
| $\rho_b$ | Bulk density | 1800 | kg m$^{-3}$ |
| $M_0$ | Volume-area scaling coefficient | 200 | m$^{3-2\gamma}$ |
| $\gamma$ | Volume-area scaling exponent | 0 | |
| $\alpha$ | Debris-flow incision law slope exponent | $2-8$ | |
| $\beta$ | Debris-flow incision law depth exponent | $0.5-3$ | |
| $D_{eff}$ | Effective grain size | 0.09 | m |
| $k_e$ | Instantaneous fluvial erodibility | $4 \cdot 10^{-14} - 6 \cdot 10^{-14}$ | m$^{5/2}$ s$^2$ kg$^{-3/2}$ |
| $U$ | Rock uplift rate | 0.5 | mm yr$^{-1}$ |
| $v_f$ | Fluid volume fraction | 0.5 | |
| $\lambda_0$ | Initial pore fluid pressure ratio | 0.9 | |
| $I_0$ | Friction factor parameter | 0.279 | |
| $\mu_2$ | Friction factor parameter | $0.625 - 0.781$ | |
| $\mu_s$ | Friction factor parameter | 0.384 | |
| $D$ | Pore pressure diffusivity | $10^{-6} - 5 \cdot 10^{-6}$ | m$^2$ s$^{-1}$ |
| $\eta$ | Viscosity of pore fluid | $40 - 80$ | Pa s |
| $k_{df}$ | Debris flow erodibility coefficient | $4 \cdot 10^{-5} - 6 \cdot 10^{-5}$ | m$^{1-\beta}$ s$^{-2}$ |

**Table G3.** Parameters used when running numerical experiments with the empirical routing model to constrain $\alpha$ and $\beta$.

| Symbol | Definition | Value | Unit |
|---|---|---|---|
| $M_0$ | Volume-area scaling coefficient | 500-3000 | $m^{3-2\gamma}$ |
| $\gamma$ | Volume-area scaling exponent | $0-1$ | |
| $\mu$ | Dynamic viscosity | $100-500$ | Pa s |
| $\tau_y$ | Yield strength | $100-600$ | Pa |
| $\alpha$ | Debris-flow incision law slope exponent | $2-8$ | |
| $\beta$ | Debris-flow incision law depth exponent | $0.5-3$ | |
| $k_{df}$ | Debris flow erodibility coefficient | $8 \cdot 10^{-5} - 2 \cdot 10^{-4}$ | $m^{1-\beta} s^{-2}$ |
| $k_e$ | Instantaneous fluvial erodibility | $4.5 \cdot 10^{-14}$ | $m^{5/2} s^2 kg^{-3/2}$ |
| $U$ | Rock uplift rate | 0.5 | $mm\ yr^{-1}$ |

**Table G4.** Parameters used in the sensitivity analysis with the process-based routing model.

| Symbol | Definition | Value | Unit |
|---|---|---|---|
| $\rho_b$ | Bulk density | 1800 | $\text{kg m}^{-3}$ |
| $M_0$ | Volume-area scaling coefficient | 200 | $\text{m}^{3-2\gamma}$ |
| $\gamma$ | Volume-area scaling exponent | 0 | |
| $\alpha$ | Debris-flow incision law slope exponent | 6 | |
| $\beta$ | Debris-flow incision law depth exponent | 1 | |
| $D_{eff}$ | Effective grain size | 0.09 | m |
| $k_e$ | Instantaneous fluvial erodibility | $2 \cdot 10^{-14} - 8 \cdot 10^{-14}$ | $\text{m}^{5/2}\,\text{s}^2\,\text{kg}^{-3/2}$ |
| $U$ | Uplift rate | $0.2 - 1$ | $\text{mm yr}^{-1}$ |
| $v_f$ | Fluid volume fraction | 0.5 | |
| $\lambda_0$ | Initial pore fluid pressure ratio | 0.9 | |
| $I_0$ | Friction factor parameter | 0.279 | |
| $\mu_2$ | Friction factor parameter | $0.532 - 0.869$ | |
| $\mu_s$ | Friction factor parameter | 0.384 | |
| $D$ | Pore pressure diffusivity | $10^{-6} - 10^{-5}$ | $\text{m}^2\,\text{s}^{-1}$ |
| $\eta$ | Viscosity of pore fluid | $30 - 90$ | Pa s |
| $k_{df}$ | Debris flow erodibility coefficient | $3 \cdot 10^{-5} - 1.2 \cdot 10^{-4}$ | $\text{m}^{1-\beta}\,\text{s}^{-2}$ |

**Table G5.** Parameters used in the sensitivity analysis with the empirical routing model.

| Symbol | Definition | Value | Unit |
|--------|-----------|-------|------|
| $M_0$ | Volume-area scaling coefficient | $1000 - 5000$ | $\mathrm{m}^{3-2\gamma}$ |
| $\gamma$ | Volume-area scaling exponent | $0 - 1$ | |
| $\mu$ | Dynamic viscosity | 100-1000 | Pa s |
| $\tau_y$ | Yield strength | 100-600 | Pa |
| $\alpha$ | Debris-flow incision law slope exponent | 6 | |
| $\beta$ | Debris-flow incision law depth exponent | 1 | |
| $k_{df}$ | Debris flow erodibility coefficient | $5 \cdot 10^{-5} - 10^{-3}$ | $\mathrm{m}^{1-\beta}\,\mathrm{s}^{-2}$ |
| $k_e$ | Instantaneous fluvial erodibility | $2 \cdot 10^{-14} - 8 \cdot 10^{-14}$ | $\mathrm{m}^{5/2}\,\mathrm{s}^2\,\mathrm{kg}^{-3/2}$ |
| $U$ | Rock uplift rate | $0.1 - 1$ | $\mathrm{mm}\,\mathrm{yr}^{-1}$ |

*Author contributions.* The initial idea for the study arose from conversations among LAM and SWM. The model code was written by LAM. WS extracted slope and area data from the San Gabriel Mountains. All authors contributed to the design of numerical experiments and interpretation of results. LAM performed the numerical experiments and wrote the paper with input and editing from all authors.

*Competing interests.* The authors declare that they have no competing interests.

*Acknowledgements.* This material is based upon work supported by the National Science Foundation under Grant No. 1951274. Jason Kean, Francis Rengers, Leslie Hsu, Ryan Gold, and Janet Carter provided comments as part of U.S. Geological Survey review. We would like to thank Alexander Densmore and an anonymous reviewer for providing reviews that improved the manuscript.

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

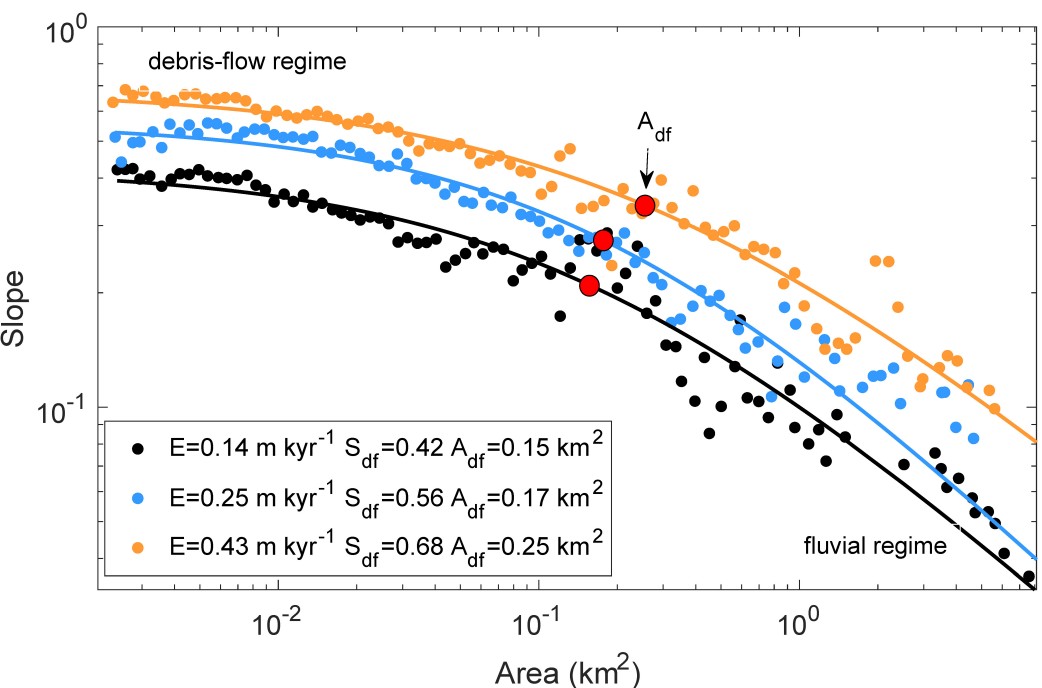

**Figure 1.** Examples of channel profiles from the San Gabriel Mountains, California, USA, along with best fit curves of the form $S = S_{df}/(1 + (A/A_{df})^{a_2})$. The locations of red circles coincide with $A_{df}$. Channel profiles were extracted from catchments with different erosion rates, $E$. Watershed outlets are located (UTM 11S) at 396887 m E 3799338 m N 396964 m E 3799133 m N (black line), 384971 m E 3799277 m N (blue line), and 417896 m E 3792642 m N (orange line). Catchment-averaged erosion rates, E, are from DiBiase et al. (2010). Slope and drainage area values were extracted from the channel network and separated into 100 logarithmically spaced bins. Binned slope was aggregated using the mean slope within each bin. The $R^2$ value for each fit is 0.95 (black), 0.97 (blue), and 0.97 (orange)

.

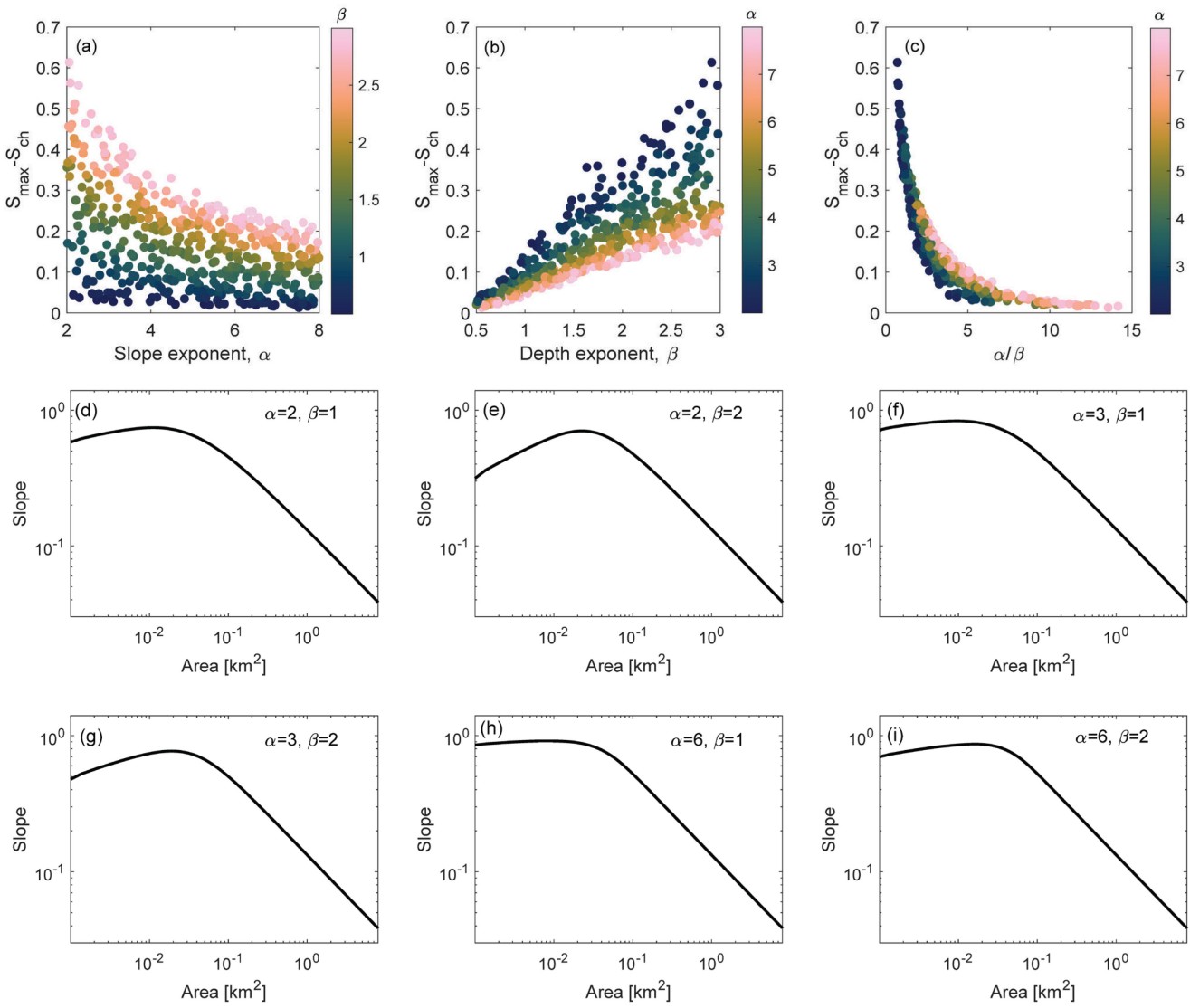

**Figure 2.** Numerical experiments using the process-based routing model to determine which slope ($\alpha$) and depth ($\beta$) exponents in the debris-flow erosion law create profiles consistent with those seen in nature. (a) The difference between the maximum slope ($S_{max}$) and the slope at the channel head ($S_{ch}$), which is the first point plotted on the profiles, generally decreases with $\alpha$. Color indicates the value of $\beta$, highlighting an increase in $S_{max} - S_{ch}$ when $\beta$ increases, and hence poor model performance in many cases with $\beta > 1$. (b) The difference between the maximum slope and the slope at the channel head generally increases with $\beta$. Color indicates the value of $\alpha$, with greater values of $\alpha$ generally leading to smaller $S_{max} - S_{ch}$ and better model performance. (c) The difference between the maximum slope and the slope at the channel head decreases rapidly with $\alpha/\beta$. (d-i) Representative profiles for different $\alpha$ and $\beta$. Substantial reductions in channel slope at small drainage areas are inconsistent with observations (e.g. equation 2).

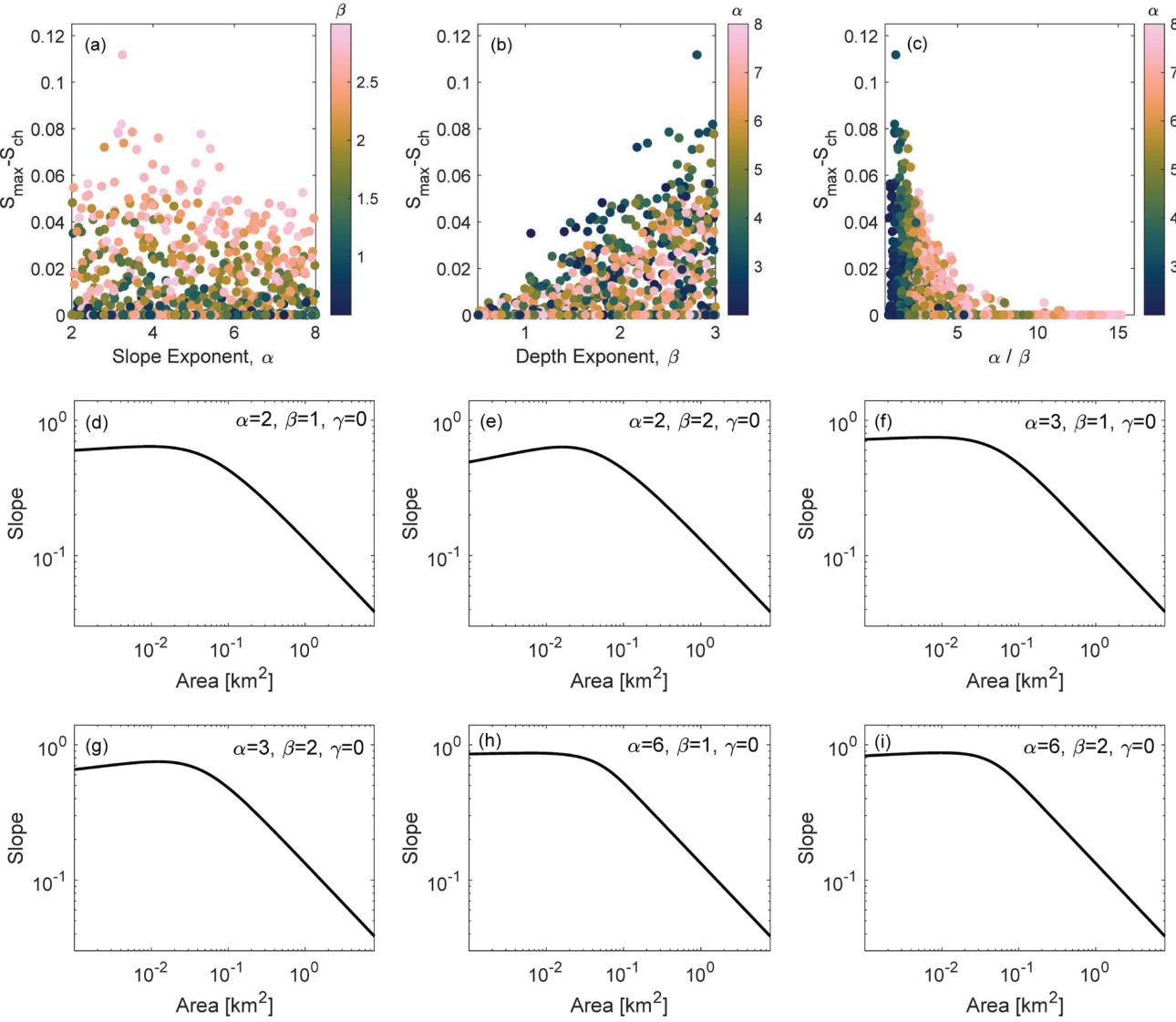

**Figure 3.** Numerical experiments using the empirical routing model to determine which slope ($\alpha$) and depth ($\beta$) exponents in the debris-flow erosion law create profiles consistent with those seen in nature. (a) The difference between the maximum slope ($S_{max}$) and the slope at the channel head ($S_{ch}$) generally decreases with $\alpha$. Color indicates the value of $\beta$, highlighting poor model performance, as measured by $S_{max} - S_{ch}$, in many cases when $\beta$ increases. (b) The difference between the maximum slope and the slope at the channel head generally increases with $\beta$. Color indicates the value of $\alpha$, with greater values of $\alpha$ generally leading to better model performance. (c) The difference between the maximum slope and the slope at the channel head decreases rapidly with $\alpha/\beta$. (d-i) Representative profiles for different $\alpha$ and $\beta$ for cases where $\gamma = 0$. Substantial reductions in channel slope at small drainage areas are inconsistent with observations (e.g. equation 2).

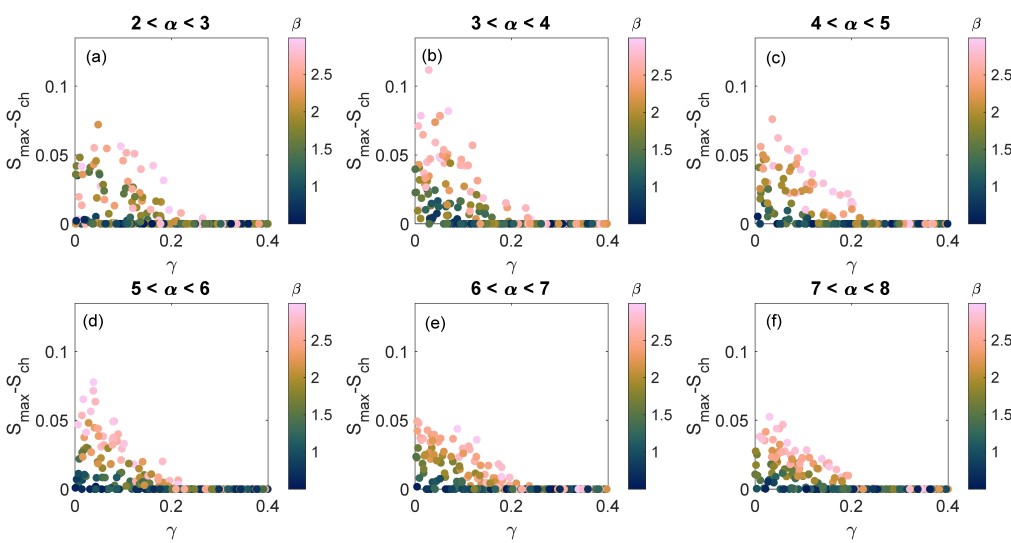

**Figure 4.** Numerical experiments using the empirical routing model that highlight the importance of the volume-area scaling exponent $\gamma$. The difference between the maximum slope ($S_{max}$) and the slope at the channel head ($S_{ch}$) generally decreases with $\alpha$, the slope exponent in the debris-flow erosion law as seen when comparing across panels a-f, and increases with $\beta$, the depth exponent in the debris-flow erosion law (as seen in marker color). Model performance, as measured by $S_{max} - S_{ch}$, is most sensitive to changes in $\alpha$ and $\beta$ when $\gamma$ is less than approximately 0.3.

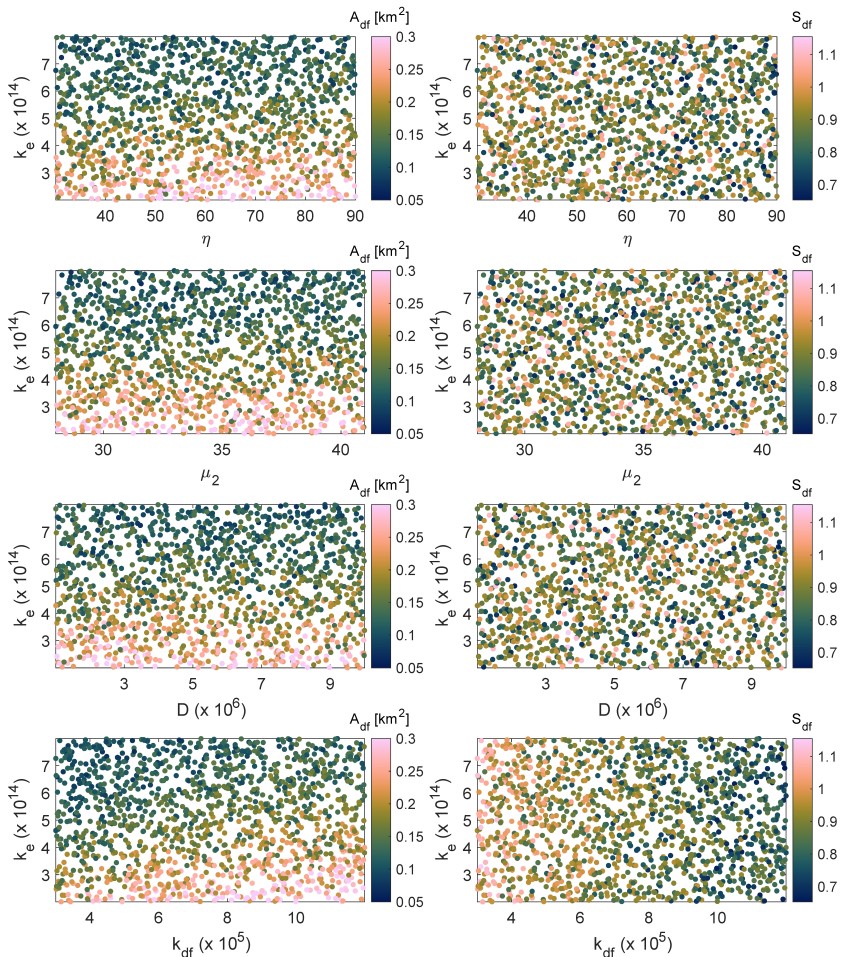

**Figure 5.** Scatter plots summarizing results of the sensitivity analysis with the process based debris-flow routing model. Sensitivity of $A_{df}$ (left column) and $S_{df}$ (right column) to particular model parameters is indicated when there is a gradient in color, whereas plots with no spatial pattern in color indicate a lack of sensitivity. The relationship between $A_{df}$, $k_e$, and different parameters related to debris-flow processes illustrate sensitivity to the debris-flow erodibility coefficient, $k_{df}$, and the instantaneous fluvial erodibility coefficient ($k_e$). The morphologic parameter $S_{df}$ (right column) is most sensitive to $k_{df}$. Parameter definitions and units can be found in Table G2.

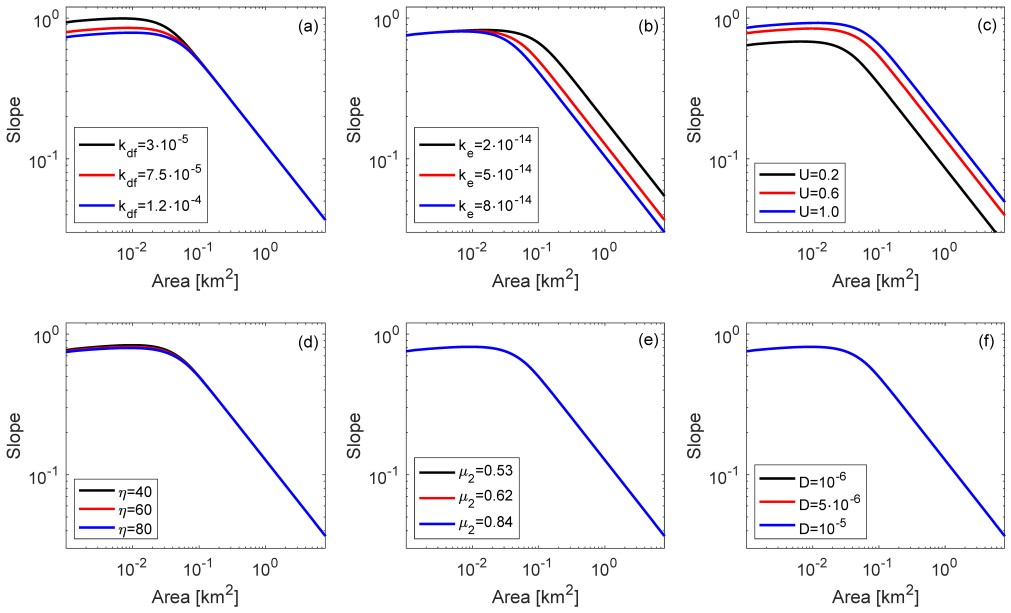

**Figure 6.** Steady-state longitudinal profiles produced by the process-based model for various parameter combinations. Variations in (a) the debris flow erodibility coefficient, $k_{df}$, (b) the instantaneous fluvial erodibility, $k_e$, and (c) rock uplift rate, $U$, drive greater changes in the channel morphology, as summarized by the relationship between channel slope and drainage area, relative to variations in (d) the viscosity of the pore fluid, $\eta$, (e) friction factor, $\mu_2$, and (f) pore pressure diffusivity, $D$. Default parameter values are: $k_{df} = 0.0001 \text{ m}^{1-\beta} \text{ s}^{-2}$; $k_e = 5 \cdot 10^{-14} \text{ m}^{5/2} \text{ s}^2 \text{ kg}^{-3/2}$; $U = 0.5 \text{ mm yr}^{-1}$; $\eta = 60 \text{ Pa s}$; $\mu_2 = 0.62$; $D = 5 \cdot 10^{-6} \text{ m}^2 \text{ s}^{-1}$.

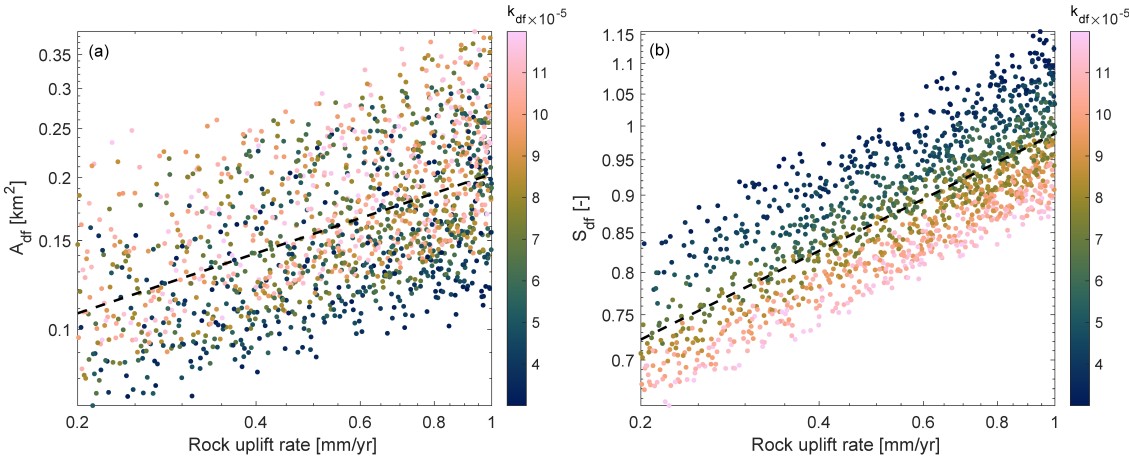

**Figure 7.** Results from the 1500 process-based model simulations in the sensitivity analysis with $\alpha = 6$ and $\beta = 1$ show a power law relationship between (a) rock uplift rate ($U$) and $A_{df}$ as well as (b) $U$ and $S_{df}$. The dashed line indicates a best fit curve. Patterns in symbol color, which represents the magnitude of the debris-flow erodibility coefficient ($k_{df}$), can be used to infer that systematic variations in $k_{df}$ with rock uplift rate, $U$, would influence the relationship between $U$ and $A_{df}$ and $U$ and $S_{df}$.

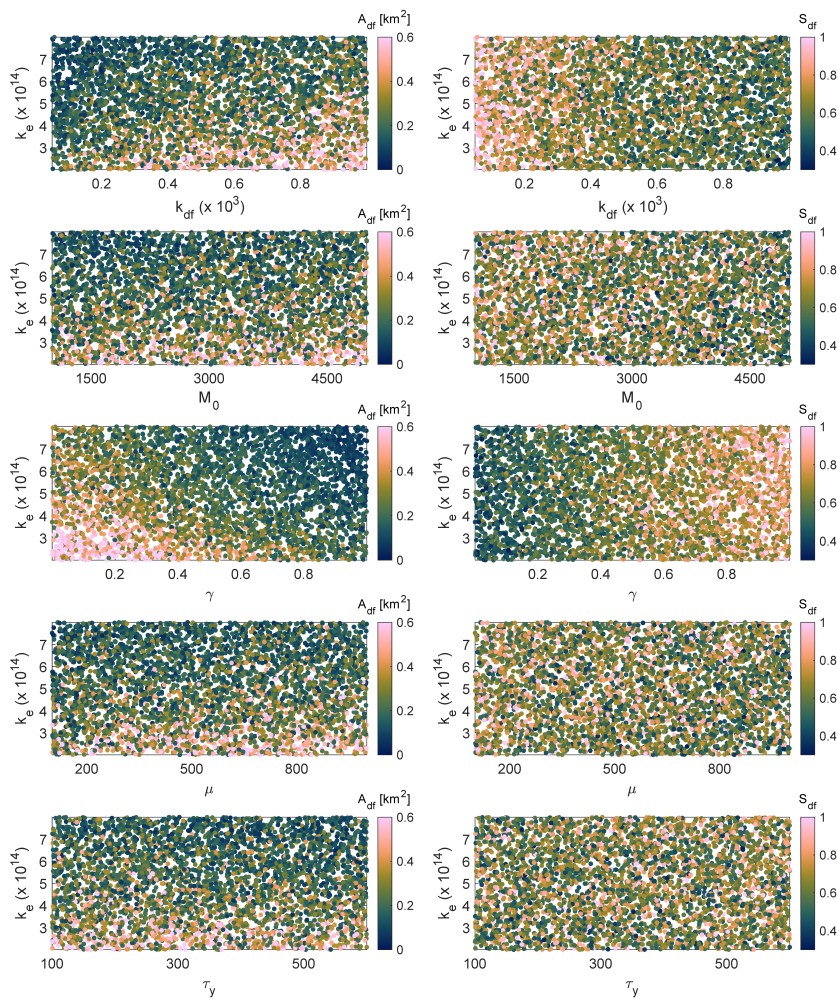

**Figure 8.** Scatter plots summarizing results of the sensitivity analysis with the empirical debris-flow routing model. The relationship between $A_{df}$, $k_e$, and different parameters related to debris-flow erosion (left panels) illustrate sensitivity to the volume-area scaling exponent ($\gamma$), debris flow erodibility coefficient, $k_{df}$, and the instantaneous fluvial erodibility coefficient ($k_e$). The morphologic parameter $S_{df}$ (right panels) is most sensitive to the volume exponent ($\gamma$) and $k_{df}$. Parameter definitions and units can be found in Table 1.

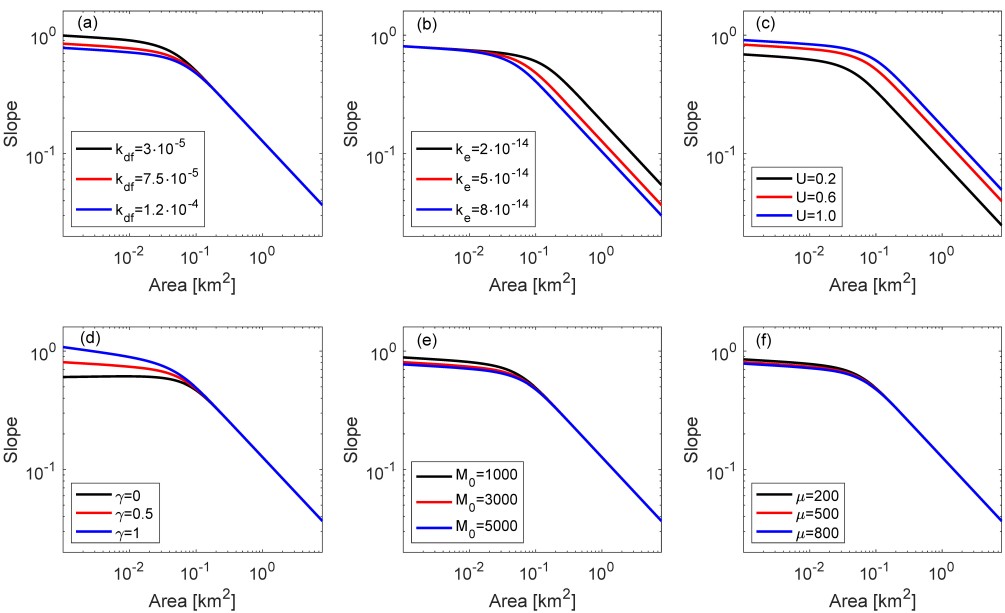

**Figure 9.** Steady-state longitudinal profiles produced by the empirical routing model for various parameter combinations. Variations in (a) the debris flow erodibility coefficient, $k_{df}$, (b) the instantaneous fluvial erodibility, $k_e$, and (c) rock uplift rate, $U$, and (d) the volume-area scaling exponent, $\gamma$, drive greater changes in the channel morphology, as summarized by the relationship between channel slope and drainage area, relative to variations in (e) the volume-area scaling coefficient, $M_0$, and (f) viscosity, $\eta$. Default parameter values are: $k_{df} = 0.0001$ m$^{1-\beta}$ s$^{-2}$; $k_e = 5 \cdot 10^{-14}$ m$^{5/2}$ s$^2$ kg$^{-3/2}$; $U = 0.5$ mm yr$^{-1}$; $\gamma = 0.5$; $M_0 = 3000$ m$^{3-2\gamma}$; $\mu = 500$ Pa s.

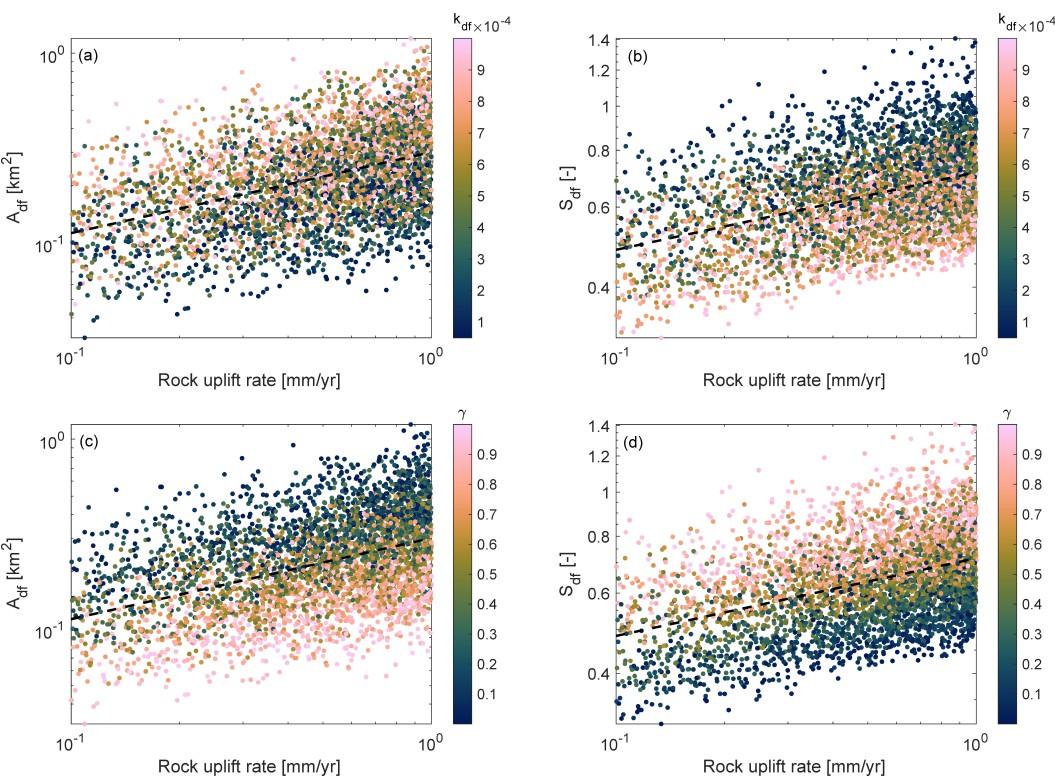

**Figure 10.** (a,b) Results from the 4000 empirical routing model simulations in the sensitivity analysis with $\alpha = 6$ and $\beta = 1$ show a power law relationship between (a,c) uplift ($U$) and $A_{df}$ and (b,d) between $U$ and $S_{df}$. Dashed lines show best fit curves. For a given rock uplift rate, changes in symbol color also highlight how $A_{df}$ and $S_{df}$ vary with $k_{df}$ such that a systematic variation of $k_{df}$ with $U$ would influence the power law fit.

**Table 1.** Model Parameters

| Symbol | Unit | Definition |
|---|---|---|
| $h$ | m | Debris-flow depth |
| $S$ | - | Channel slope |
| $A$ | m$^2$ | Upstream drainage area |
| $w$ | m | Channel width |
| $k_w$ | m$^{1/2b}$ | Width-area scaling coefficient |
| $b$ | - | Width-area scaling exponent |
| $Q$ | m$^3$ s$^{-1}$ | Debris-flow discharge |
| $c_1$ | - | Discharge coefficient |
| $c_2$ | - | Discharge exponent |
| $M$ | m$^3$ | Debris-flow volume |
| $M_0$ | m$^{3-2\gamma}$ | Volume-area scaling coefficient |
| $\gamma$ | - | Volume-area scaling exponent |
| $\mu$ | Pa s | Dynamic viscosity |
| $\tau_y$ | Pa | Yield strength |
| $\rho_b$ | kg m$^{-3}$ | Bulk density |
| $\alpha$ | - | Debris-flow incision law slope exponent |
| $\beta$ | - | Debris-flow incision law depth exponent |
| $k_{df}$ | m$^{1-\beta}$ s$^{-2}$ | Debris-flow erodibility coefficient |
| $t_p$ | s | Debris-flow passage time |
| $\Theta$ | - | Threshold factor in debris-flow incision law |
| $\Phi$ | - | Threshold factor in debris-flow incision law |
| $m_s$ | - | Stream power law area exponent |
| $n_s$ | - | Stream power law slope exponent |
| $k_e$ | m$^{5/2}$ s$^2$ kg$^{-3/2}$ | Instantaneous fluvial erodibility |
| $U$ | m yr$^{-1}$ | Rock uplift rate |

**Table 2.** PAWN Sensitivity Indices: Process-based Model

| Parameter | Definition | Sensitivity Index ($A_{df}$) | Sensitivity Index ($S_{df}$) |
|---|---|---|---|
| $\eta$ | Viscosity of pore fluid | 0.06 | 0.09 |
| $\mu_2$ | Friction factor parameter | 0.05 | 0.07 |
| $D$ | Pore pressure diffusivity | 0.08 | 0.05 |
| $k_{df}$ | Debris flow erodibility coefficient | 0.13 | 0.25 |
| $k_e$ | Instantaneous fluvial erodibility | 0.39 | 0.08 |
| $U$ | Rock uplift rate | 0.23 | 0.35 |

**Table 3.** PAWN Sensitivity Indices: Empirical Model

| Parameter | Definition | Sensitivity Index ($A_{df}$) | Sensitivity Index ($S_{df}$) |
|---|---|---|---|
| $M_0$ | Volume-area scaling coefficient | 0.06 | 0.06 |
| $\gamma$ | Volume-area scaling exponent | 0.28 | 0.29 |
| $\mu$ | Dynamic viscosity | 0.04 | 0.07 |
| $\tau_y$ | Yield strength | 0.06 | 0.04 |
| $k_{df}$ | Debris flow erodibility coefficient | 0.12 | 0.19 |
| $k_e$ | Instantaneous fluvial erodibility | 0.18 | 0.03 |
| $U$ | Rock uplift rate | 0.15 | 0.13 |