# Peer review of "Steady-state forms of channel profiles shaped by debris-flow and fluvial processes"

_Earth Surface Dynamics, 2022_

## Author Comment (AC1)

We would like to thank both reviewers for their thorough reviews of our manuscript. Here, we briefly summarize a few of the more substantial revisions we have made to address reviewer comments:

1. Addition of a new subsection to the Methods section that focuses on a simplified analytical solution for slope in the debris-flow-dominated portion of the channel profile. As mentioned by the reviewer, this provides additional insight into how profile shape is affected by b, alpha, beta, and gamma. The analytical solution also helps motivate the numerical model experiments.
2. Modifications to the abstract and introduction to better communicate the scope of the study.
3. Addition of an analysis of channel profile morphology in the San Gabriel Mountains that quantifies changes in slope between Adf and the channel head.
4. Addition of an analysis that quantifies the relationship between channel width and drainage area in the San Gabriel Mountains.
5. Modified the equation used to estimate shear stress in the empirical debris-flow routing model.
6. Addition of a figure showing the effects of a spatially variable debris-flow frequency on channel profile morphology when using the process-based routing model. This is a simple way to parameterize an overall increase in sediment transported by debris flows as drainage area increases in the process-based model.
7. Addition of a quantitative global sensitivity analysis using the PAWN method to rank the relative importance of different model parameters.

Below, we indicate reviewer comments in italics, followed by our response.

**Reviewer 1**

*The sharp slope inflection in the area-slope relationship observed for low-order channels has long been documented in worldwide topographies, and has been explained as due to dominant debris flow erosion. However, the controls and magnitude of channel head erosion by debris flows have been addressed in a limited manner: direct field measurements of erosion are almost non-existent, and models are also lacking. McGuire and co-authors attempt to address this second issue by proposing two separate models based on several parameters, and then explore the sensitivity of these models to their different parameters. The main constraint to validating their model is the qualitative observation that the slope of the upper part of the channels increases slowly but continuously to the head of the channel.*

*The subject matter of this research is appropriate for the journal Earth Surface Dynamics, and the proposed approaches could represent an interesting first step in the direction of quantifying the role of debris flows in landscape evolution. Unfortunately, the way the study is designed and the paper is written does not really accomplish this goal. I would actually*

*suggest that the authors completely reconsider the organization of their paper, including a modification (or deletion) of the process-based model to include the first-order components that are not considered and yet have a major impact on the shape of the inflection discussed in the preamble.*

*First, as long as the only proposed validating constraint is the observation that "the slope continues to increase or remain constant as drainage area decreases…", the whole process-based model, as it is proposed, should be rejected, even if for very high value of alfa this increase becomes so attenuated that it could be confused (in the sense of the R2 criterion used by the authors) with a uniform slope not depending on A. This left me a little confused!*

*A preliminary simplified analytic exploration of the empirical model (which is very briefly and incompletely done on line 381) would have permitted the authors to identify the important ingredients for the model, by indicating quite clearly what controls the area-slope relationship along the channel heads (i.e. for A < Adf). Considering the erosion equation (eq. 4), and equation (13), the expression for "tp" (eq 14) and assuming as a first approximation that capital theta is uniform and close to 1 (shear stress >> tauy) along the head channel reaches, we can derive such relation at steady state.*

*S depends on uplift and drainage area according to a power relation: S proportional to A^N, with*

*N = -(beta/3.(gamma.c2 – b) + gamma.(1-c2))/(alfa – beta/3)*

*The sign of this exponent N controls the slope of the S=f(A) relationship for channel heads. For explored values of alfa >= 2, the denominator is always positive so that the main controlling parameter appears to be the numerator, and among others the sign of (gamma.c2 – b) modulated by beta. We immediately see that if this term is positive then the slope is constantly decreasing downstream. In contrast if this term is negative then the downstream slope increase observed on fig. 2 and 3 becomes possible. If c2 is fixed, the most fundamental parameters are gamma and b. In other words for gamma =0, for the empirical as well as probably for the process based model, the downstream decrease in erosive efficiency (for a uniform channel slope) is due to the channel widening (parameter b) that induces a decrease of the flow thickness h. If gamma is not zero, then there is a critical value for which the slope trend reverses (for gamma larger than b/c2 according to the authors, but in fact for gamma larger than beta.b/3/(1 + (beta /3-1).c2) ).*

*Once these two main influences on the sign of the slope of S=f(A) have been identified, the one due to the widening of the channel and the one due to the increase in the volume of debris flowing downstream, we must ask ourselves:*

1. *if they correspond to a reality;*
2. *if so, if they are well taken into account by the model:*

*To the first question, one can notice that the parameter b is given for the fluvial domain, but that its implicit transposition to the debris flow domain, as done in this article, has no theoretical or empirical basis. Insofar as this debris flow eroded domain behaves differently from the fluvial domain, the transposition is difficult to justify. That an important trend in the model results is controlled by a parameter that is unknown and uncalibrated is quite problematic. Furthermore, as described below, the assumption of w>>h (although not made explicit in the paper) and of a rectangular channel is another problem associated with how w = f(A, M) can be injected into the equations. Regarding the increase in the volume or frequency of debris flows downstream, one need only look at Fig. 2 of Stock and Dietrich (2003) to see that the number of debris flow sources in the contributing basin of a given point will increase downstream, more or less in proportion to the area drained. Taking this increase into account is essential to any model looking at the long-term evolution of the channel profile.*

*To the second question, we can notice that the process-based model does not include this essential element. This poses a double problem: firstly if we want to compare the performances of the two models, the boundary conditions must be the same (the addition of sediment downstream can be seen as a boundary condition), and more importantly if it is an essential element to the results, it must be implemented (below, I suggest to the authors a quick way to take it into account without needing to modify the core of their equations). In other words, the process-based model is unnecessarily complicated in some aspects while it does not include first order elements.*

*Given these two deficiencies, it seems to me that the proposed models are for the moment of little use to the community in that some important ingredients are missing and in that these models do not clearly pass a validation or refutation criterion, so that it is impossible to say whether or not these models are suitable to reproduce reality.*

*Moreover, the architecture of the paper should be modified. It seems to me that a clearer and more rational approach from the point of view of the construction of a physical model would be schematically the following:*

*1. What are we trying to demonstrate or test? A spatially variable model of instantaneous erosion? An erosion model representative of the long term geometry at equilibrium (if this notion means anything on slopes affected by landslides)?*

R: We aim to construct a flexible framework to incorporate debris-flow erosion into a model designed to simulate the evolution of channel longitudinal profiles over geologic time scales. We focus our modeling efforts on channel profiles that have attained an approximate steady state. A landscape can still be said to be in an approximate (dynamic) equilibrium while being affected by stochastic processes such as landsliding or debris flow. Topography may still be approximately constant when averaged over a certain time/spatial scale. As for a rigorous validation of a debris-flow erosion law, that is beyond the scope of our study. We

present a family of debris-flow erosion laws and propose a framework that can be used to incorporate these, and other similar debris-flow erosion laws, into a landform evolution model. We demonstrate that a subset of the proposed family of debris-flow erosion laws produces channel profiles that are inconsistent with observations from debris-flow-dominated landscapes. In our response to more specific comments below, we highlight changes made to the abstract, introduction, and discussion that we hope will help clarify the objectives and conclusions of the study.

We acknowledge, as pointed out by the reviewer, that additional analyses are needed to quantify relationships between channel width and drainage area in debris-flow-dominated landscapes and that debris-flow frequency may vary spatially with interesting implications for channel form. In terms of initial model development, it is critical to have a flexible framework that allows for improvements to be made (e.g. to width-area scaling or volume-area scaling, etc) as new data emerge. The framework that we present, particularly when using the empirical debris-flow routing method, is flexible in terms of its ability to parameterize many of the processes mentioned above. Although some of the parameterizations are poorly constrained at present, results help identify and prioritize observational targets that would be most critical to advancing our ability to incorporate debris-flow erosion into landform evolution models. We emphasize this in the revised abstract: "Results improve our ability to interpret topographic signals within steep channel networks and identify observational targets critical for constraining a debris-flow incision law."

*2. What are the constraining observables to validate or invalidate the models? In the submitted study, if I understand correctly the only constraining observable, presented just in a qualitative way is the fact that in general, past the fluvial/debris flows transition the slope continues to increase going towards the source. This constraint being unique, it is essential to be clear on this constraint. Is this observation general? Or is it just observed for 3 drainages in the San Gabriel Mtns (fig. 1)? It would be helpful to offer a mini-synthesis of observations made on this topic in the literature. And to add quantitative criteria (e.g. the ratio between the slope at the source (Sch) and the slope at the transition (S(A=Adf)), or another criterion quantifying whether the slope remains stable or continues to increase above the transition A=Adf). On this point of quantification it is essential to know if the slope continues to increase as suggested by the authors. If so, as said before, this systematically disqualifies the process-based model that predicts an increase in the downstream slope between Ach and Adf regardless of the values of alfa and beta.*

R: The curve (eq. 1 in the revised text) suggested by Stock and Dietrich (2003) was developed following an analysis of channel longitudinal profiles across a range of geographic areas. We therefore interpret it as representing a general pattern seen in many locations, not only the San Gabriel Mountains. Since some of our model parameters are set based on past work in the San Gabriel Mountains, we conducted an analysis in the San Gabriel Mountains and found that slope-area data for 30 out of 31 watersheds indicate an

increase or constant slope as drainage area decreases between Adf and the channel head. Slope decreases in one instance by roughly 0.03. The following text has been added to the methods section:

"Equation 1, which was formulated by Stock and Dietrich (2003) as part of an analysis of channel morphology across a range of geographic areas, suggests that channel slope increases or remains approximately constant as drainage area decreases. We performed an analysis of 31 channel longitudinal profiles in the San Gabriel Mountains, USA, to determine the frequency with which channel slope decreased as drainage area decreased. The San Gabriel Mountains were chosen for this analysis because some of our model parameter choices are based on previous studies in this mountain range and topography is in an approximate steady state (DiBiase et al., 2012). We extracted channel profiles for a subset of catchments with 10Be catchment-averaged erosion rates (DiBiase et al., 2010), where we eliminated catchments with signs of disequilibrium such as knickpoints. In 30 of the 31 catchments, slope increased or remained approximately constant as drainage area decreased. In one catchment, there was a difference of 0.03 between the maximum slope along the channel profile and the top of channel profile (Fig. S3)."

As for these data disqualifying the process-based model, we do not think results demonstrate this in a general sense. The process-based model does not account for downstream changes in debris-flow volume. Therefore, these observations would only serve to disqualify a model that uses the process-based routing approach and does not include downstream changes in debris-flow volume. The same could be said for a model that does not account for downstream increases in debris-flow volume and uses the empirical routing approach.

Given the lack of constraints used to assess model performance in this study, we also modified elements of the discussion to emphasize that we do not think that reproducing the curve given by equation 1 is sufficient to validate or uniquely determine the form of a debris-flow erosion law:

"Results indicate that many members within the proposed family of debris-flow incision laws, as formulated by equations 5 and 6, produce channel profiles that are consistent with observations from natural landscapes (Figs. 2, 3). This is true within a wide range of the parameter space explored here, including for a range of gamma that covers the variability observed across several different geographic regions reported by Santi and Morandi (2013). Data and numerical experiments presented here are not capable of differentiating among these potential debris-flow incision laws, although cases where alpha<3 and/or beta>2 generally performed poorly (Figs. 2, 3). Additional work is needed to formulate and test a debris-flow incision law, including incision laws not restricted to the form of equation 5 (e.g. Stock and Dietrich, 2006)."

However, we also would like to emphasize that the analytical solution and results of numerical experiments both point to observational targets for future studies (i.e. improved characterization of channel morphometry in debris-flow-dominated terrain) that would be critical for constraining a debris-flow incision law. We mention this in the revised abstract: "Results...identify observational targets critical for constraining a debris-flow incision law."

*3. To build the model, one needs to keep the essential elements (as for a Taylor expansion, do we keep all the details at order 1 (there is no point in keeping terms of order 2 if all the terms of order 1 are not kept). In the absence of a theoretical framework allowing to make this choice, one can at least define, given the points 1 and 2, what are the elements of the model that it is essential to keep. I understand that it can be complex to introduce into the equation (6) an aggregation term (M increasing downstream) of the sediments (and of its momentum) during downstream transport, but it is on the other hand extremely easy to conceive just a multiplicative term in the frequency of passage of the debris flows at a given point, which increases according to the drained area (and to the number of upstream talwegs likely to generate debris flows departures) _ this is equivalent to introduce a "kdf" that would depend linearly (or not) on A.*

R: Thank you for this suggestion. The modification that you recommended was straightforward to implement with the process-based routing model. We explored the effects of scaling the debris flow erodibility coefficient (i.e. a multiplicative term in the frequency of passage of the debris flows at a given point) such that the total volume passing a given point in the channel profile is related to drainage area. A figure summarizing these results has been added to the Appendix. Results are similar to those obtained when using the empirical routing model and scaling debris-flow volume with drainage area.

*4. Propose in particular for a simplified model like the empirical model here a first simplified analytical resolution to predict the main trends. In the present study, given this analysis highlighting the role of gamma as a parameter conditioning the increase or reduction of A downstream, the phenomenon carried by the gamma parameter cannot be neglected or dismissed. It must be taken into account. At this stage, the authors in their study should have resumed their model, added this aggregation to the process-based model, and proposed a new model (i.e. discard the old model which can be considered as a first draft) and only talk to us about this last model.*

R: We agree that it would be useful to include a more complete description of the simplified analytical solution. We have added a section titled "Simplified Analytical Solution" that describes the assumptions needed to derive an analytical solution and some of the basic insights that we can gain from examining it. This section helps to motivate the numerical experiments. However, we still see value in presenting the process-based model. A key point for including the process-based model is to demonstrate that, under circumstances where debris-flow volume does not change in the downstream direction, the empirical and process-based models yield qualitatively similar results. This increases confidence in some

of the simplifying assumptions of the empirical routing model. There is also value in presenting the process-based model because future studies could add in entrainment terms to allow debris-flow volume to change as the flow moves downstream or otherwise parameterize increases in the total sediment volume transported by debris flows as a function of drainage area (e.g. by prescribing a debris-flow frequency that increases with drainage area as suggested above). We better motivate our logic for comparing channel profiles produced when using the process-based and empirical routing models with the addition of the following text in the methods section:

"This comparison is limited, as described in the following sections in more detail, to cases where downstream changes in debris-flow volume are assumed to be negligible. While we acknowledge this is not likely to be true in many natural settings (Santi et al., 2008; Santi and Morandi, 2013; Schürch et al., 2011), examining this end-member case allows for the most direct comparison between channel profiles produced by the model when using these two different debris-flow routing methods."

*5. Verify or deepen these first conclusions using the numerical simulation. For the "empirical" model, the simulation will allow us to take into account the capital theta term and to have an analysis based on an unapproximated solution.*

R: We have revised the methods section to include an exploration of an analytical solution for channel morphology in the upper network. We agree that this helps set up the numerical experiments and refine/focus discussion points.

*6. Possibly propose a more advanced model if the empirical model does not allow to account for the observables.*

R: We do not think this is needed, though results indicate that additional constraints are necessary to better test and validate debris-flow erosion laws.

*Other issues :*

- *I found in several equations some problems with the dimensions that are not respected; because I did not check everything in detail, I encourage the authors to recheck all the equations. There is also a vagueness about the volume of debris flow and how it is introduced for the process-based model. To be corrected.*

R: Thank you for pointing this out. We have corrected errors in equations and parameter units as noted below in response to specific comments.

- *Why are the ranges of exploration of the parameters external to the model ($M_0$, $k_{df}$, $k_e$ ...) not the same for the two models? If we want to compare the performances and predictions of the two models, it seems to me obvious to explore the same ranges of values.*

R: We have made adjustments to use the same parameter ranges in some cases, namely for the instantaneous fluvial erodibility (ke) in the sensitivity analysis. If we were solving the same set of equations with different numerical methods, it would be critical to be consistent with parameter ranges. In this case, the equations used to determine bulk flow properties are sufficiently different that we are not attempting to directly compare the two models (i.e. we do not directly compare modeled flow depths or flow velocities and we do not quantitively compare channel profiles). Our goal is to explore large portions of the parameter spaces associated with models that are constructed using the process-based and empirical routing approaches to assess which erosion laws, out of the proposed family of potential erosion laws, produce channel profiles consistent with observations. The parameter spaces differ depending on which routing model (process-based or empirical) is used.

- *If we want to model the landscape, then it is required to be conservative with respect to the sediments. This is indirectly addressed in the discussion through the coupling between U and kdf , but it must be done more rigorously (especially since it is simple to do). For instance (lines 339, 356), the choice of the relations between U and kdf is totally arbitrary, and is not even the same between the two models (this choice is not trivial because it will condition the subset of orange points and the slope of the relation Sdf=f(U) for this). For example, for the process based model, increasing the uplift rate by a factor 10 leads, given the chosen coefficients within the inequality, to vary kdf by on average a factor 2. In theory, and excluding a small proportion of material exported by other processes ( wet ravelling, subfluvial process? ), an increase of erosion rate by a factor 10 should lead to an increase of the debris flow frequency by a factor 10 (assuming that their volume remains constant).*

R: We do not think that the model needs to assume that debris-flow volume and/or frequency increase with rock uplift rate in order to conserve sediment. By not prescribing a relationship, we are implicitly assuming that more sediment is eroded by fluvial processes as rock uplift rate increases. To match trends between rock uplift rate and Adf (or Sdf), we agree that it would be necessary to justify a relationship between rock uplift rate, debris-flow frequency, and debris-flow volume. We are actively working on this problem (Struble, W., L. McGuire, S. McCoy, and K. Barnhart. "Quantifying the role of debris flows on steepland evolution." In AGU Fall Meeting Abstracts, vol. 2021, EP25F-1383. 2021), but it is beyond the scope of this study.

- *It seems to me that in order to reproduce a long term geometry, introducing a temporal distribution of debris flows could be necessary. Indeed, the authors rely for the fluvial part on the relation proposed by Lague (2014). It seems to me that one of the main conclusions of this study is that the exponents of the law E=f(S,A) depend strongly on the distribution of floods because the instantaneous incision law includes a threshold (tauC) below which erosion is zero. For debris flows,*

*since there is also a threshold for the motion onset or efficiency of erosion (tau_y), one can anticipate that the resultant of the mean law will be sensitive to the combination of a threshold and an event distribution with small events traveling little distance because h and tau will be small), and that this may impact the position of Adf, as well as the shape of the transition between the two domains which will be more gradual. In other words, it is again a matter of trying to be consistent: as the problem (and in particular the transition zone) depends on the law of river incision downstream and debris flow upstream, it is important to include the same level of detail in the models on both sides.*

R: We agree that it will be interesting to explore the role of temporally varying debris-flow properties, but we do not think this is essential for a starting point. In this study, we focus on presenting a framework that will help facilitate that work in the future. We have added text to clarify our assumption that debris flow properties do not change over time:

Changed in line 119 to "We propose a general formulation that can be used to estimate the erosion rate attributable to a debris flow, Edf, at a point on the landscape, given information about the bulk properties of the flow. In this work, we assume that bulk properties of a debris flow for a given landscape position, do not change. In other words, debris flow erosion is driven over time by repeatedly routing the same debris flow over the landscape."

We also acknowledge that future work could use the framework we are proposing to account for debris flows that have properties (e.g. volume, yield strength) described by distributions rather than fixed values.

- *I would suggest adding a schematic graph describing the process-based model, and (in appendix?) one or more results of the propagation of a debris flow downstream as simulated by the process-based model.*

R: Thank you for this suggestion. We have added a figure to the Appendix that shows the propagation of a debris flow as it is routed down a channel profile with the process-based model as well as the magnitude of erosion attributable to debris-flow and fluvial processes along the channel profile at steady state.

*Figure 1 caption : add « (eq. 17) » to link the equation to the text. Indicate the projection system (UTM zone ??). I tried to look at the location of these points but it does not correspond to a drainage basin that can be unambiguously identified)*

R: We have updated points for watershed outlets and clarified that these points are in UTM zone 11.

*Figure 1: add the P value for ach fit*

R: We have added the $R^2$ values to the caption for each fit. "The $R^2$ value for each fit is 0.95 (black), 0.97 (blue), and 0.97 (orange)."

*Line 68: would it be possible to indicate another reference, i.e. other than this PhD thesis that cannot be easily accessed ?*

R: The only reference for this work is the PhD thesis. We have minimized use of this citation in the revised text.

*Line 111: those values seem to me quite arbitrary. Why this choice? In addition, the ratio ms/ns=0.6 seems a bit high compared to classical curvature parameter values of 0.4-0.5.*

R: The ms and ns values are the result of parameter choices for the fluvial incision law that are consistent with the San Gabriel Mountains as described in the Appendix. We have revised the text to clarify:

"These parameter choices result in $m_s=1.4$ and $n_s=2.33$. Complete details on parameter choices for the stream power model are given in Appendix A1."

*Line 114: I would suggest to provide a number to this equation, and to discuss more at length the choice of the parameters, and in the discussion the implications of this choice.*

R: The implications are minor for this study because there is nothing that is parameterized within the debris flow models as a function of channel length. The relationship between area and downstream distance mostly affects the length of the model domain. See figure below for examples of a modeled channel profile at steady state using different exponents and coefficients in this relationship.

[Figure]

*Line 126 and Eq (3): where this equation coming from? It needs to be explained and justified. This relation is not detailed at all and refers to a thesis that is not readily available online and to a work that has not been peer reviewed. It is impossible for me to judge its relevance in these conditions, and it seems essential to me to publish beforehand or to include in this paper the developments proposed in this PhD thesis.*

R: We removed the citation to McCoy (2012) and have added motivation for this particular family of erosion laws: "Motivated by observations that debris-flow erosion rates scale with bulk inertial stress (Hsu et al., 2008, 2014), a function of shear rate, and that grain-scale bed-impact force distributions scale with flow depth (McCoy et al., 2013), it is reasonable to postulate an erosion law that includes debris-flow depth and velocity. Since steady granular flows down inclined planes of increasing angles show a monotonically increasing relationship between slope angle and velocity (Silbert et al., 2001), slope may serve as a proxy for velocity. Here, we define…."

We also want to emphasize that we view the use of this erosion law as a starting point for exploring the model framework presented here and have added some text to reflect that thinking: "We use this family of erosion laws to begin exploring the model framework we propose here. The model is designed in such a way that it would be straightforward to insert alternative erosion laws in the future."

*Line 127: for reasons given above (and to justify/discuss the sensitivity to uplift rate), it would be more appropriate to display Fdf explicitly instead of hiding it in kdf.*

R: Since we primarily assume debris flow frequency does not vary in this study, we decided to continue to subsume this into kdf. We define kdf and its relationship with debris flow frequency when it is introduced.

*Line 142 (eq. 4): this equation is not homogenous. Either some terms are missing (like the frequency of debris flows), or the kdf units (as given in table B2 to B6) is incorrect*

R: Units should be m^(1-beta) s^(-2).

*Line 153: "in a rectangular channel" This is a major hypothesis. As much a river channel constrained by its banks or in a canyon can possibly present a rectangular section, as much an ephemeral channel head presents, for what I saw in the nature, a rather widened or prismatic shape. This choice was made for simplicity I assume, but it would be necessary to discuss the adequacy of this assumption and in the discussion whether having a wider channel would change the results.*

R: The relationship between drainage area and channel morphology will affect results and it is partly for this reason that more detailed analyses will be needed in specific landscapes in order to use the framework proposed here to better constrain and test debris-flow incision laws. The updated examination of the analytical solution in the methods section helps to elucidate the role of channel width in controlling how slope varies with drainage area near the channel head.

*Line 156 (eq. 6): this equation contains several errors. The 3rd term on the left hand side is not homogeneous. I assume it is rather gz.h^2 I assume that gx (1rst term on the right hand side) has to be replaced by the projection of the weight onto the channel sloping direction*

*(otherwise, gx=0); and similarly, gz has to be replaced by the projection of the weight onto the direction normal to the sloping channel bed.*

R: Correct, the term you identified should read "g_z h^2". g_x denotes the component of gravity in the downslope direction while g_z denotes the component of gravity in the direction normal to the bed.

*Line 187: "introducing this effect … is beyond the scope of this study". This sentence is quite paradoxical: why do you decide not to incorporate this effect in the process based model, and to do so in the empirical model? If you want to compare the performance of the two models, then you need to consider equivalent boundary conditions and hypothesis on flow volumes.*

*Again I presume that incorporating this effect in the eq. (6) is uneasy. However, one can easily play with the frequency of debris flows to introduce this dependence (linear or by a power relation with an exponent gamma' between 0 and 1) to the drainage area. In that case, you should do it similarly for the two models (i.e. not consider the relation M=M0.A^gamma for the empirical model, and saying/demonstrating that you will capture the two effects with only one process)*

R: Yes, we can modify the frequency of debris flows in both models. We have added some results where we use the process-based debris flow routing model and a spatially variable debris-flow frequency in order to mimic the effects of increasing debris-flow volume with increasing drainage area (Figure D1).

More generally, we have tried to revise text to emphasize that we are not attempting to compare the process-based routing model and the empirical routing model for every modeling scenario. We chose an end-member modeling scenario (i.e. no changes in debris-flow volume with drainage area) that could be simulated with both debris-flow routing models. We compare the results between the two for this end-member case and focus on whether or not the models agree or disagree on the range of exponents in the debris flow erosion law that result in channel profiles that are consistent with equation 2. The following text has been added to the methods section to reflect this reasoning:

"This comparison is limited, as described in the following sections in more detail, to cases where downstream changes in debris-flow volume are assumed to be negligible. While we acknowledge this is not likely to be true in many natural settings (Santi et al., 2008; Santi and Morandi, 2013; Schürch et al., 2011), examining this end-member case allows for the most direct comparison between channel profiles produced by the model when using these two different debris-flow routing methods."

*Line 194-198: this part was not clear to me until I realized that there is no distribution of the volumes of debris flow but always the same one running through the channel. Did I understand correctly?*

R: Correct, there is not a distribution of debris-flow volumes in the current work. We have added the following text to the methods section and hope this helps to clarify: "In this work, we assume that bulk properties of a debris flow for a given landscape position, do not change. In other words, debris flow erosion is driven over time by repeatedly routing the same debris flow over the landscape."

*Line 206: this equation seems to me oversimplified: first one should use the hydraulic radius instead of h, except if one can demonstrate that w>>h; second for steep slopes, S should be replaced by sin(theta) with theta the slope angle. The more exact equation should be first written and then the potential simplification justified*

R: Thank you for pointing this out. We now compute the shear stress using the sin of the slope angle and the hydraulic radius and have revised the model code and the description in the text.

*Line 210: "we specify debris flow volume at each grid cell". Do you mean "passing through each grid cell "?*

R: Yes, changed to "...passing through each grid cell...".

*Line 211: how A is expressed ? in m2 ?*

R: The expression from Santi and Morandi (2013) assumes that A is given in $km^2$ so the factor of $10^{-6}$ is needed to convert units.

*Line 225 (eq 13): where is  the 10^-6 factor in front of A. Are A units now km2?*

R: Thank you for pointing this out. We have added the factor of $10^{-6}$ to the relevant equations.

*Line 305: "some parameter combinations …": some ? No, ALL parameter combinations according to fig.2 for which Smax is always strictly larger than Sch. '*

R: Changed to "all parameter combinations".

*Line 308: "… is inconsistent with observations that indicate slope continues to increase or remain constant as drainage area decreases…" . In other words, given above remarks, the whole process-based model should be rejected … as long as it does not include a downstream increase of the  volume or of the frequency of debris flows.*

R: We agree that this indicates that a process-based model with a constant debris-flow volume and constant debris-flow frequency produces longitudinal profiles that are inconsistent with most observations from debris-flow dominated landscapes. The same can be said for a model that uses the empirical debris flow routing approach.

*Lines 331, 332: those conclusions are a bit obvious (except on the relation between U and Adf). No need of doing numerical simulations for this.*

R: We agree that some of these are intuitive results but we would still like to briefly mention them. We have also expanded our sensitivity analysis to include a ranking of the model parameters in terms of their relative importance for controlling Adf and Sdf. These results can help guide future studies that hope to use channel morphology to constrain/test debris-flow erosion laws or studies that employ measures of channel morphology in debris flow-dominated terrain as a proxy for erosion rate (e.g. Penserini et al., 2017).

*Line 364: replace kdf by gamma within the inequality. Again the choice of a coefficient 0.5 seems quite arbitrary. In addition, and in contrast with the DF frequency, it remains unclear to me why this gamma parameter should be modified with U.*

R: The imposed relationship between U and gamma was arbitrary. This analysis was designed to illustrate one of the implications of Adf being sensitive to gamma. Penserini et al. (2017) demonstrate how it can be informative to use a1 as a morphologic proxy for erosion rate in steady state watersheds, in a manner analogous to how channel steepness is used to estimate spatial variations in erosion rate. However, if a1 or Adf are to be helpful as morphologic proxies for erosion rate, it is beneficial to understand if/how these metrics may be sensitive to other factors that may vary spatially or with rock uplift rate. For example, one considers spatial variations in climate and rock type when using channel steepness to infer erosion rates. What are the most important factors to control for if attempting to use a1 or Adf to infer erosion rate? To better address this question, we removed the text tagged here by the reviewer and replaced figures 7 and 10 with new figures. Rather than highlighting changes in the relationship between U and Adf and U and Sdf for specific relationships between kdf and U (or gamma and U), we aim to better highlight how correlations between kdf and U or gamma and U may generally influence U-Adf and U-Sdf relationships.

*Line 376: "data and numerical experiments presented here are not capable of differentiating… although cases where alfa < 3 and beta > 2 generally perform poorly". The authors are quite honest and objective in this sentence. They should start from this sentence, instead of introducing the section insisting on the fact that McCoy's (2012) model with alfa=6 and beta =1   perform well. It seems to me that present study does not permit to reject this model, but neither does it validate it.*

R: Thank you for the suggestion. We agree that the analyses presented do not allow us to validate or reject the model proposed by McCoy et al. (2012). We have revised the discussion section so that it now starts with the following:

"Results indicate that many members within the proposed family of debris-flow incision laws, as formulated by equations 5 and 6, produce channel profiles that are consistent with observations from natural landscapes (Figs. 2, 3). This is true within a wide range of the

parameter space explored here, including for a range of gamma that covers the variability observed across several different geographic regions reported by Santi and Morandi (2013). Data and numerical experiments presented here are not capable of differentiating among these potential debris-flow incision laws, although cases where alpha<3 and/or beta>2 generally performed poorly (Figs. 2, 3). Additional work is needed to formulate and test a debris-flow incision law, including incision laws not restricted to the form of equation 5 (e.g. Stock and Dietrich, 2006)."

*Line 407-408: This is obvious: from the moment when, by definition of equation (17), Sdf is defined where the slope becomes constant and deviates from the fluvial relation S=f(A), it goes without saying that Sdf becomes disconnected from any parameterization of the fluvial law (except if ms is close to 0).*

R: We agree that this is intuitive but we would still like to briefly mention it since fluvial erosion is still an active process along the entire channel profile.

*Line 418: is Penserini et al. (2017) the only paper that tested the relation between channel head slope and uplift rate? Have the authors checked the literature in the whole US, Italy, Taiwan, Himalaya-Tibet, etc ?*

R: We think that the Penserini et al. (2017) study is the most relevant for this discussion topic since they specifically examine variations between Sdf and erosion rate. Definitions of channel head slope vary among studies and channel head slope may not correspond with Sdf. It should also be kept in mind that robust quantification of Sdf or Adf requires high resolution DEMs and linkages between these metrics and erosion rate are also limited by availability of erosion rate estimates (e.g. from 10Be). We are not aware of other equivalent studies to reference at this point in the text.

*Line 423: ok for the regression in fig. 10b but in contrast the difference in fig. 7b does not seem major between orange and blue points.*

R: We have modified Figures 7 and 10. Instead of showing line plots of power law relationships under different assumptions (e.g. kdf scales linearly with U), we color all data points by kdf or gamma. This allows us to visually illustrate how relationships between kdf and U or gamma and U would influence the relationships between Adf and E or Sdf and E.

*Lines 445-446: these lines are redundant with lines 418. The whole discussion should be condensed on these points.*

R: We have moved the text formerly starting at line 418 to the following section ("Tectonics from debris-flow processes and topography") to help streamline the discussion and emphasize how erosion rate and channel morphology may be connected in steep, quasi-steady channels.

*Line 457: "the channel width scaling, b, may … exert control over the long term channel". I would rather say that this unconstrained parameter has a primary role in the fact that the slope increases downstream until ~Adf instead of slightly decreasing.*

R: Width-area scaling in debris-flow dominated landscapes will be important to quantify as future studies continue to make progress on incorporating debris-flow incision into landform evolution models. We have added the following text: "Analytical (i.e. equation 25) and numerical (Figs. 5, 8) model results indicate that relationships between drainage area and debris-flow volume as well as drainage area and channel width, in particular, play key roles in determining the morphology of the debris-flow-dominated channels."

*Lines 476, 487: "the landscape evolution model presented here …". This model cannot be called a landscape evolution model because it is 1D and, above all, does not conserve water or sediments. Introducing progressive aggregation of larger sediment supply as we go donwstream in order to respect steady state erosion of the landscape would be the minimum. The introduction of gradual aggregation of larger sediment fluxes as one moves downstream (whatever it I achieved playing with M or Fdf) in order to respect steady-state landscape erosion would be the minimum requirement in that direction.*

R: Based on a similar comment from Reviewer 2, we now say **"**landform evolution model" in what was formerly line 476 and "a model for channel profile evolution" in line 487**.**

As also noted above, we do not think that the model needs to assume that debris-flow volume and/or frequency increase with rock uplift rate in order to conserve sediment. By not prescribing a relationship, we are implicitly assuming that more sediment is eroded by fluvial processes as rock uplift rate increases.

*Line 487: "demonstrate" should be replaced by "propose"*

R: Changed to "propose."

*Appendix A: Given that many parameters are jut arbitrary (for example Deff) , I don't see the point to describe this section since it is already done in Lague (2014). But if the authors prefer to keep that section for the reader, then the instantaneous incision law should be explicitly described or written.*

R: We prefer to keep this section so that notation can be defined in this work and the relationship between key variables is clear to the reader. The instantaneous incision law is defined in the last sentence of Appendix A as $E_f = K A^{m_s} S^{n_s}$.

*Line 503: if Rc is a runoff, it should have units (m or m/yr)*

R: Changed to "0.28 m".

*Line 504: the equation should be provided with a number (A1 ?) and t should be subscripted into "kt".*

R: Done.

*Table B3: why M0 is not varied among the different parameters?*

R: We assumed a constant debris-flow volume in these simulations.

*Tables B3 and B5: Problem of units for ke*

R: Units changed to be consistent with other tables.

*Table B3 and B4: I would suggest putting in a different table the fluvial and forcing parameters, which are common to the two models (U, M0, ke)*

R: Given how text is structured in the main document, we prefer the current format where parameters used in each numerical experiment are contained in separate tables.

*Table 5: U is given as constant whereas it is supposed to vary over a certain range*

R: Changed to 0.2-1.

**Reviewer 2**

*This is an interesting and timely manuscript that seeks to develop an approach for modelling the combined effects of debris-flow and fluvial incision on river long profiles. Debris flows have long been recognised as important agents of erosion and sediment transport in many mountain catchments, but they have typically been left out of attempts to model channel erosion or landscape evolution. The authors have taken some initial steps toward that goal. The manuscript is highly relevant to the journal and will, I think, be of interest to the journal readership.*

*The manuscript is well-written and well-presented overall, but I do have some comments on the text and figures. Most of these are fairly minor and should be easy for the authors to address. One more substantive comment is that I was surprised to see that neither the process-based nor the empirical model conserve mass, and only the empirical model considers flow volume variation downstream (although this is imposed as being monotonic). Work in the Illgraben catchment and elsewhere has shown that flows can both lose and gain volume downstream, so while the imposed rule from Santi and Morandi (2013) is certainly a place to start, it would be good to see a little more context around its usage. More importantly, I wondered about comparing the two models given that one assumes the flow volume is uniform downstream and the other does not.*

R: We have added some additional context around the usage of the volume-area scaling relationship proposed by Santi and Morandi (2013). When we first mention downstream changes in debris flow volume, we have added:

"In nature, we expect debris-flow volume to vary with drainage area as sediment is entrained and deposited along the runout path (Santi and Morandi, 2013; Schürch et al., 2011; Santi et al., 2008),..."

Also, later in the text when we introduce the relationship between volume and drainage area proposed by Santi and Morandi (2013):

"Debris-flow volume may not increase monotonically along the runout path (Schurch et al., 2011), but the formulation proposed by Santi and Morandi (2013) provides a useful starting point for a general parameterization of downstream variations in debris-flow volume, especially since there are data from a range of geographic regions to fit such a relationship. For example, Santi and Morandi (2013) demonstrate that …"

Thank you for pointing out that the motivation for comparing the process-based and empirical models was not clear. We have added the following text to the methods section to help clarify that we only compare the two models for an end-member case where downstream changes in debris-flow volume are neglected:

"This comparison is limited, as described in the following sections in more detail, to cases where downstream changes in debris-flow volume are assumed to be negligible. While we acknowledge this is not likely to be true in many natural settings (Santi et al., 2008; Santi and Morandi, 2013; Schürch et al., 2011), examining this end-member case allows for the most direct comparison between channel profiles produced by the model when using these two different debris-flow routing methods."

In addition, to address reviewer comments about comparisons between the two routing approaches when one assumes downstream changes in debris-flow volume and the other does not, we have added results of simulations where we parameterize a downstream increase in sediment transported by debris flows when using the process-based model. We accomplish this increase in sediment transported by debris flows by changing the frequency of debris flows as a function of drainage area rather than changing the volume as a function of drainage area. See Appendix D.

*The introduction, while clear and pretty easy to follow, lays out a few different motivations for the work that don't necessarily all track through the rest of the manuscript. I think this could be streamlined and focused on what the authors are actually doing here. For example, the mention of the need to 'identify robust topographic signatures of debris-flow erosion' isn't something that is addressed here – instead, they are taking a single measure (slope increases monotonically upstream but at a decreasing rate, as in Fig 1) as that signature. It would have been good to be more clear about this up front. Fig 1 is cited as an*

*example of this, but it's not clear how widespread that morphology is; a parameter A_df and an equation are introduced in the caption but aren't actually described in the text until p. 10, which could be confusing for the reader. Elsewhere in the intro, there is a goal (lines 86-88) which isn't really clearly motivated at that point in the manuscript, and a separate set of objectives (91-97), but then there are two other goals on lines 239-242 that overlap with the third and fourth objectives. I'd encourage the authors to restructure the introduction to keep the focus on what they are going to do here (e.g., while I agree that flow frequency is critical as mentioned on lines 78-80, that's not something they address), and to motivate the model comparison that is at the heart of this manuscript.*

R: Thank you for these suggestions to improve the introduction and clarify the objectives. We have made the following changes:

1. We have moved text from the methods section to the introduction so that equations 1 and 2 in the revised manuscript were formerly equations 16 and 17 in the original version. We hope this helps introduce Figure 1.
2. Moving equations 16 and 17 to now become equations 1 and 2 also allows us to mention early on how we use equations 1 and 2 as criteria to assess model performance ("We examine the extent to which different erosion laws are capable of reproducing the relationship between slope and drainage area, as captured by equation 2, that has been observed in steep, debris-flow prone landscapes and interpreted as a topographic signature of debris flows.")
3. Deleted the following text: "The goal of this comparison is to assess the extent to which the empirical approach, which may be more readily integrated into two-dimensional (2d) landscape evolution models, yields results that are consistent with the process-based approach."
4. We have modified the abstract to better highlight the model comparison and need to develop methods for estimating spatial variations in bulk debris-flow properties (e.g. "To quantify the impact of debris-flow erosion and steep channel network form, it is first necessary to develop methods to estimate spatial variations in bulk debris-flow properties (e.g. flow depth, velocity) throughout the channel network that can be integrated into landscape evolution models. Here, we propose two methods to estimate spatial variations in bulk debris-flow properties (e.g. flow depth, velocity) along the length of a channel profile. We incorporate both methods into a model designed to simulate the evolution of longitudinal channel profiles that evolve in response to debris-flow and fluvial processes. To explore this model framework, we propose a general family of debris-flow erosion laws where erosion rate is a function of debris-flow depth and channel slope.")
5. We heavily edited the last paragraph of the introduction where we briefly outline the remainder of the paper.

*Some more specific comments by line number:*

*line 2: 'rates and spatial patterns of landscape evolution by debris flows' is ambiguous – should this be '…of erosion by debris flows'?*

R: Changed to "Debris flows regularly traverse bedrock channels that dissect steep landscapes, but our understanding of bedrock erosion by debris flows on steepland morphology is still rudimentary."

*36-40: these sentences are written as if these are two different concepts, but aren't they equivalent?*

R: Yes, the results we are referring to from DiBiase et al. (2012) and Penserini et al. (2017) both support the conceptual model proposed by Stock and Dietrich (2003) that the transition from a nearly linear debris-flow dominated long-profile to a concave-up fluvial-dominated long-profile migrates out to larger drainage areas as the rock uplift rate increases. However, DiBiase et al. (2012) and Penserini et al. (2017) used different methods and quantified changes in channel morphology in different ways so we wanted to be careful not to directly equate their findings. We have rephrased these ideas to try to better highlight how DiBiase et al. (2012) and Penserini et al. (2017) support the conceptual model proposed by Stock and Dietrich (2003): "Past work demonstrates that the length of the channel network upstream of the debris-flow fluvial transition zone, which we roughly associate with $A_{df}$ , increases with erosion rate in two landscapes where debris flows are known to regularly traverse steep channels, namely the San Gabriel Mountains (DiBiase et al., 2012) and the Oregon Coast Range (Penserini et al., 2017). These results from DiBiase et al. (2012) and Penserini et al. (2017) are consistent with the conceptual model proposed by Stock and Dietrich (2003) where the …**"**

*45: that identification has (apparently) already been made in line 34. This is repeated again in lines 49-50*

R: Thank you for pointing this out. We also agree with your earlier assessment that this statement does not set up the broader objectives of this study. We have modified it as follows:

"These findings underscore the need to develop a quantitative framework that can be used to explore topographic signatures generated by debris-flow erosion, assess the sensitivity of topographic signatures to climatic and tectonic forcing, and ultimately interpret these signatures to gain process-based insights about the evolution of steep landscapes. In particular, there is a need to understand the relative importance of fluvial and debris-flow processes in setting the location and form of the morphologic transition associated with $A_{df}$."

*50: here and throughout the manuscript, I would suggest using 'rock uplift rate' consistently. There are also places where U is variously used for 'uplift rate' or just 'uplift', and again I think this needs to be made consistent*

R: Changed here and throughout the manuscript to "rock uplift rate."

*71-73: while I agree, I think this sentence is also missing the idea that this will happen over multiple flows which themselves are drawn from distributions of volume and flow properties, if we are interested in landscape evolution. This sentence could be read as being about properties in a single flow that's traversing the landscape*

R: Changed to "The utility of these relationships in a landscape evolution model, however, requires tractable simulation of the spatial and temporal variability in the properties of individual debris flows (e.g. depth, velocity, shear rate) throughout the channel network and integration of the effects of numerous debris flow events on channel evolution over geologic time scales."

*79: I think this should read 'For example, the frequency…has been shown to be a key factor'*

R: Changed to "For example, the frequency at which …"

*130: given that flow volumes and properties are not constant from flow to flow but follow distributions, I'm not sure what is meant by a 'representative' flow. I think it's really important to be explicit that eqn 3 is defined for a single flow, and that it is being applied over a series of flows that are assumed (rightly or wrongly) to be identical. That's a really restrictive assumption and I think it needs to be made more obvious. The authors are later clear about how future work could use these distributions (lines 235-237), which is great.*

R: Changed in line 119 to "We propose a general formulation that can be used to estimate the erosion rate attributable to a debris flow, Edf, at a point on the landscape, given information about the bulk properties of the flow. In this work, we assume that bulk properties of a debris flow for a given landscape position, do not change. In other words, debris flow erosion is driven over time by repeatedly routing the same debris flow over the landscape."

*140: again this is assuming that h is constant for a given channel location in a single flow. And given that this is a representative value for h rather than the true flow thickness over time (as used in eqn 3) I wondered whether a different symbol would make sense…*

R: We use "h" throughout to refer to debris-flow depth. The "h" used here is the same as the "h" determined from the empirical debris-flow routing method.

*167: this is potentially confusing because later D is allowed to vary – not sure why a single value is specified here*

R: We have deleted the value specified for D here since it does vary.

*184-187: our work at the Illgraben (Schuerch et al. 2011 Geology) documented flow volume variations with distance downstream, both positive and negative as flows traversed the lower part of the catchment and the fan*

R: We agree that debris flows may gain or lose volume along their travel path and that volume changes do not need to be monotonic. Thank you for pointing us to this work at the Illgraben, which illustrates this point. We have added references to Schürch et al. (2011) and modified the text in two places. When we first mention downstream changes in debris flow volume, we have added:

"In nature, we expect debris-flow volume to vary with drainage area as sediment is entrained and deposited along the runout path (Santi and Morandi, 2013; Schürch et al., 2011; Santi et al., 2008),..."

Also, later in the text when we introduce the relationship between volume and drainage area proposed by Santi and Morandi (2013):

"Debris flow volume may not increase monotonically along the runout path (Schurch et al., 2011), but the formulation proposed by Santi and Morandi (2013) provides a useful starting point for a general parameterization of downstream variations in debris-flow volume, especially since there are data from a range of geographic regions to fit such a relationship. For example, Santi and Morandi (2013) demonstrate that …"

*187-189: in other words, mass is not conserved, right?*

R: Mass is conserved in the debris flow component of the model and in the greater landform evolution model. This sentence is referring specifically to the mass of the debris flow. The total debris flow mass does not change as the flow moves downstream. At some point, the flow stops and deposits all of its sediment. We do not track this sediment in the model. The implicit assumption we make is that any debris-flow sediment deposited in the channel is rapidly eroded by fluvial processes and transported out of the model domain.

The sediment added to a debris flow by bedrock erosion will be negligible compared to the volume of the debris flow that is sourced from other locations in the watershed. For example, if a debris flow uniformly erodes 1 mm of bedrock along an 8 km travel path in a channel that is, on average, 1 m in width, this will result in the addition of 8 m^3 of sediment to the debris flow. Debris flows in the model are substantially larger so neglecting sediment sourced from bedrock erosion is reasonable. We have added the following sentence to clarify:

"Regardless of which routing approach is used, however, we do not explicitly account for rock mass incorporated into the debris flow originating from bedrock incision. This sediment volume would be negligible compared to the total debris-flow volume."

*195-199: I think I followed this, but it could perhaps be more clearly explained*

R: We expanded our description of how the time stepping works, including addition of the following text:

"The term…is not updated with each time step in the landscape evolution model. Small changes in topography, such as may occur during a single time step of the landscape evolution model, will not substantially affect flow mobility or spatial variations in flow depth along the runout path…"

*250-266: this partly repeats text in the intro, but this is actually clearer and introduces eqn 17 which has already been shown. I think this text should be merged with the intro so that Fig 1 can be better understood*

R: The equation proposed by Stock and Dietrich (2003) to represent the shape of the slope-area curve has been moved to the introduction and related text in this section has been merged with that in the introduction as suggested.

*274: up to here the tense has been present (We assess… we compute), but here it changes to past. This should be kept consistent*

R: Changed throughout the numerical experiments section to past tense.

*272-273: I don't disagree with this criterion… but it's kind of hidden here, despite the fact that this becomes the primary way in which the authors accept or reject model runs. I think this needs to be highlighted (perhaps in the intro where they are describing what they are trying to match)*

R: Thank you for this suggestion. The following text has been added to the introduction: "We propose a general family of debris-flow erosion laws, with erosion rate being a function of debris-flow depth and channel slope, to illustrate how the proposed framework may be used to help constrain a debris-flow erosion law and to explore model sensitivity. We examine the extent to which different erosion laws are capable of reproducing the relationship between slope and drainage area, as captured by equation 2, that has been observed in steep, debris-flow prone landscapes and interpreted as a topographic signature of debris flows."

*277: parameter k_e is introduced here but eqn 2 is in the form of K – it's not obvious where k_e has come from. A reference to appendix A might help*

R: We added a reference to appendix A.

*288: I think I missed the flow frequency – is there a flow every year? Or every timestep?*

R: Debris-flow erosion occurs continuously through time, as does fluvial erosion. Debris-flow frequency is subsumed into the kdf parameter in the debris flow incision law. An increase or decrease in debris-flow frequency could be parameterized by increasing or decreasing kdf.

*327: 'of the steady-state channel profiles' – I'm not sure if this is referring to the model results or to observations*

R: Here, we are referring to model results. Changed to "Two defining characteristics of the simulated steady-state channel profiles…"

*328: I think this should be 'minimum drainage area'?*

R: Changed to "...and the minimum drainage area at which…".

*336: 'increases with rock uplift rate'. More broadly, while I agree with the overall sense of the argument here and this is certainly a reasonable supposition (all else being equal), I'm not sure what is gained by exploring a single, arbitrary relationship between U and k_df. This might just need some more contextual information. Flow frequency should also depend very strongly on where you are in the catchment…*

R: Our motivation for exploring a single relationship between U and k_df was to illustrate that it can change the relationship between E and Adf. This is important for future applications where one may want to explore the use of Adf as a proxy for catchment-averaged erosion rate. We have modified figures 7 and 10 to better highlight how a relationship between U and kdf would affect the relationship between E and Adf. In particular, we no longer assign a single relationship between U and kdf and instead plot Adf as a function of U, with data points colored by kdf.

*343: I don't think the 'e.g.' makes sense here, because k_df is explicitly defined as the product of an erodibility and flow frequency*

R: Changed from "...(e.g. debris-flow frequency, erodibility)..." to "..., which is related to debris-flow frequency and bedrock erodibility,..."

*350: 'rock uplift rate U'*

R: Changed to "rock uplift rate, U, …."

*375: I'm not sure what is meant by 'infrequent instances' – is this referring to individual numerical experiments in the sensitivity tests, or localised parts of the profile within a single experiment, or…?*

R: This text has been deleted as substantial changes were made to this section.

*381: if E_df needs to decrease slightly with A, why does that imply that h must increase with A? That seems to run counter to eqn 4*

R: Much of this section has been reworked now that we show and examine the analytical solution in greater detail, beginning in the methods section.

*383: reference to flow discharge is potentially confusing here, because gamma has been defined in terms of downstream changes in flow volume, not discharge. I agree that they might be related, but that relationship isn't necessarily simple (or the same for all flows)*

R: This text has been removed as we have revised and expanded upon our original discussion of the simplified analytical solution for channel slope.

*398: 'rock uplift rate'*

R: Changed to "rock uplift rate" as suggested.

*406: I'm a little confused by this leading statement – isn't this conceptual model what has been assumed? In which case, how can the results be seen as supporting this conceptual model? I suppose what the authors are saying is that there are parameter sets for which this assumed model form can reproduce aspects of observed long profiles – that's a more restrictive statement (which they make elsewhere). That doesn't rule out other conceptual models which might also reproduce those long profile aspects, of course*

R: The results of simulations from our channel profile evolution model do not necessarily need to support this conceptual model for channel evolution. It is possible, for example, that the debris-flow incision term would have played only a minor role in which case the conceptual model (that both debris flow and fluvial processes play important roles) would have been incorrect. In contrast, we found that both fluvial and debris-flow processes controlled Adf, which is an important metric for quantifying the general morphology of channel profiles. We have replaced this statement with one that is more specific and sets up the discussion in the rest of the paragraph:

"Model results help clarify the roles played by debris-flow and fluvial erosion processes in setting longitudinal profile form in the upper channel network."

*472: because we are considering landscape evolution, then flow volume will change not just by sediment entrainment from the bed and banks but also by bedrock erosion*

R: Yes, but entrainment from bedrock into any individual debris flow will be negligible compared to the overall debris-flow volume.

*479: 'where changes in flow volume can be neglected' – I'm not sure when that would ever be the case, given that real flows have continuously-varying downstream lag rates which*

*can be both positive and negative (and can both erode and deposit within a single channel cross section)*

R: Here, we are re-iterating the assumptions of the process-based model in its current state and agree that the inability to account for debris-flow volume changes is limiting.

*482-485: that's true, although there isn't much mention throughout of t_p in eqn 4 – this seems like a pretty big unknown if the goal is to model channel erosion over kyr-Myr time scales. I guess this comes back to the idea of a 'representative' debris flow, which elsewhere the authors also refer to as a 'characteristic' flow, and what that means*

R: When using the empirical routing model t_p can still be cast as a function of drainage area and slope. We agree though that there would still be a lot to work needed when moving this methodology into a 2d landscape evolution model.

*487: I'm not sure I'd call this a landscape evolution model – it's really a model of channel profile evolution*

R: Changed to "a model for channel profile evolution".

*Appendix A: D_eff has units of m which should be repeated here*

R: Added "m".

*Fig 1: the equation for the best-fit curves is going to be confusing for the reader because it doesn't show up again in the text until p. 10 – as noted above I suggest moving that material to the intro.*

R: Thank you for this suggestion. We have moved the relevant text and equations into the introduction.

*Fig 4: the panels here seem to show the same thing as Fig 2 and don't match the caption.*

R: Thank you for pointing this out. An older version of this figure was included by mistake and that is why the caption did not match the results shown on the figure.

*Fig 5,8: I think it would be useful to include the units for each parameter on the axis labels*

R: We refer the reader to the relevant table in the caption for units and parameter definitions given the limited room on the figure for longer axis labels.

*Fig 6,9: again it would be useful to include the units for each parameter, either in the legend or caption*

R: We have added units to the captions.

*Fig 7,10: 'Uplift' should be 'Rock uplift rate' on the x-axis, and the units are given in m/kyr here but in m/yr in most other parts of the manuscript; the same comment applies to Fig 10.*

R: Changed to "Rock uplift rate" and units changed to mm/yr.

---

## Author Response (AR2)

L. McGuire and his co-authors present a revised and improved manuscript of their investigation on channel profiles shaped by debris flow and fluvial processes. The authors provide a careful replies and revisions in response to my comments and suggestions. They largely modified the text, and added new figures in the main body and mostly in supplementary information.

One of the major points that I had underlined in my previous review concerned the fact that in most of the proposed models the slope increases downstream in the part dominated by debris flows whereas the natural examples of the San Gabriel Mtns showed the opposite. The authors now recognise this misfit more widely, and discuss it more extensively, in particular by insisting on the role of the volume aggregation parameter or the frequency of downstream debris flows. This discussion is now partly supported by a simplified analytical formulation. This does not necessarily make the models more convincing but it does help to better define the limits of these models, to better highlight the controlling parameters and the important aspects to be implemented for future developments. From that scientific point of view, I have no further concerns that should prevent the publication of this study.

These modifications and improvements to the scientific approach have nevertheless been made slightly at the expense of clarity and fluidity of the reading. The addition of remarks, caveats and arguments ahead of the discussion tends to disrupt the rhythm of the reading in some places.

Some adjustments would make the reading more fluid. As a matter of examples:

Line 23-24: this precision appears a bit like a digression in the state of the art. Either the authors should modify the sentence to insert it better, or move this sentence further on in the manuscript.

**R: We moved the text to a later paragraph in the introduction. It now reads:**

**"….where A denotes upstream drainage area, and Sdf, a1, and a2 are empirical coefficients (Fig. 1). Here, we use the term channel in a general sense to refer to an axis of concentrated erosion along valley bottoms, but which may or may not reside within banks made of deposited sediment. The coefficient…."**

Line 50-55: the two sentences seem to me to provide more or less the same message. This could be simplified.

**R: We have chosen to keep the current text because there are differences in methodology and conclusions in the cited studies that would be challenging to communicate as well in a single sentence.**

The fact that one can vary an equivalent of "gamma" for the process-based routing model is mentioned in sentences scattered throughout the text without this being clearly explained except in the appendix D. The text would gain in clarity (e.g. the sentence on line 208 "An increase in slope with drainage area near the channel head, however, is not an inevitable consequence of using the process-based routing model (Appendix D)" is rather cryptic) if the main text had more explanation on this point. I understand the overall approach because I was aware of this point from the start. However, I am not sure that all readers will be able to follow the line of the argumentation.

**R: We have added a sentence (in italics below) to the methods section that helps set up the results in Appendix D. "When using the process-based debris-flow routing model, we assume that debris-flow volume is fixed and does not change along the flow path, although we do explore the effects of spatial variations in debris-flow volume with the empirical routing approach described later. *In addition, we***

*perform a set of numerical experiments with the process-based routing model where we scale debris-flow frequency with drainage area to account for an increase in the total volume of sediment transported by debris flows as drainage area increases.* **Regardless of which routing approach is used…"**

Just as the content of Appendix D is poorly explained and little used in the main text, the same applies to Appendix B: the values of the coefficients "kw" and "b" seem to fall out of the air, whereas Appendix B and Fig. B1 are there precisely to allow empirical values to be proposed. The authors should build on this. Otherwise, Fig. B1 is of little use and might as well not be included.

**R: We made a modification where Figure B1 is cited to indicate that this figure provides data that help constrain kw and b:**

**"Motivated by the geomorphic importance of debris flows in the San Gabriel Mountains (Lavé and Burbank, 2004), parameters related to channel geometry, including kw and b that are related to width-area scaling (Fig B1), and ….."**

The use of Sfit(A0) in fig. B2 is confusing. Is it different or the same as Sch ?

**R: Sfit(A0) is different from Sch. We do not attempt to precisely quantify the location of the channel head from the actual data.**

On the other hand, if the discussion reads relatively well, I find that the justification (in the introduction and presentation of the models) and especially the correspondence (in the discussion) between the two models are not sufficiently explored. Even if the authors recognise that it is complicated to compare them (and describe how one could move from one to the other) because of the very different parameters, there is probably more that could be done to enlighten the reader on the relative pertinence of and correspondence between these models. In the end, one is left with the impression of having explored two models in parallel and not knowing which one to use, or with the impression that the model at the bottom does not really matter: basically, you have just to push the alfa exponent to high values so as to guarantee a quasi-constant slope in the debris flow domain, and then to modulate this slope by playing on Kdf.

**R: We provide guidance on when to use the two models in the discussion section: "The process-based routing model may be best suited for modeling 1d channel profiles where changes in flow volume can be neglected and debris-flow constituents are sufficiently well known to allow for estimates of the model parameters, thereby minimizing the number of numerical experiments needed to characterize model behavior. The empirical debris-flow routing algorithm provides an efficient framework for investigating the effects of different debris-flow bulking relationships and exploring large parameter spaces"**

Table 1: the threshold factor used in equation (5) should be added

**R: Done.**

Figure 1: in the equation of the figure caption, the exponent "p" must be replaced by "a2". In addition, optimal "a2" value could be indicated for each profile.

**R: p has been changed to a2 in the caption of figure 1.**

Figure 2: It is a bit strange to indicate on the figure "gamma=0" knowing that this parameter does not appear in the process-based routing model formalism. The reader should have read first appendix D (fig. D1) to understand the relationship that can be established with gamma. This is part of the vagueness mentioned above about gamma and the process-based routing model.

**R: We have removed "gamma=0" from Figure 2. In the text, we state that setting gamma equal to zero is equivalent to neglecting changes in debris flow volume with drainage area.**

Figures 2, 3, 4, 5 and in sup info: it is a personal appreciation, but I found the blue to red color scale, used in the initial draft, easier to read.

**R: We have opted to keep the current color scheme.**

Figure 4: This figure has been added compared to the initial version. It is useful to show at which value of gamma the maximum slope of the channel is located at the channel head and not farther downstream. Nevertheless, given the presence of figure B2 with channel data from the San Gabriel, it would have been more interesting to represent S(A0) - S(Adf), to see for which values of gamma, we will find values between 0 and 0.2 as on the histogram of fig. B2

**R: We have kept S_max-S_ch in Figure 4 since we use this quantity as a model performance metric.**